# Enhancing Protein-Protein Interaction Prediction with Hierarchical Motif-based Multimodal Protein Embedding

**Zaifei YANG** [1]   **Samuel Ping-Man Choi** [2]   **James Kwok** [1]

## Abstract

Protein-protein interactions (PPIs) are essential for many biological processes. However, existing PPI prediction approaches suffer from two major limitations: they overlook the hierarchical organization of proteins, particularly meso-scale motifs that critically regulate PPIs, and fail to effectively integrate sequence, structure, and function modalities. To address these limitations, we propose MMM-PPI, a *Hierarchical Motif-based* *MultiModal protein Encoder for PPI Prediction* that constructs PPI embeddings in a **bottom-up multi-modal** manner across three scales. At the **micro-scale**, we encode three modal residue features; at the **meso-scale**, a novel multimodal motif encoder aggregates residues into spatially-informed motif embeddings; at the **macro-scale**, a multimodal protein encoder integrates motifs into protein embeddings by jointly modeling motif importance and inter-modal correlations. The pre-trained encoder can be used off-the-shelf for large-scale PPI prediction. Extensive experiments on multiple PPI datasets show that MMM-PPI outperforms state-of-the-art multi-label PPI prediction models, particularly under challenging data partitions and limited data scenarios. Codes are in https://github.com/yzf-code/MMM-PPI.

## 1. Introduction

Protein-protein interactions (PPIs) (Phizicky & Fields, 1995; Hu et al., 2021) are fundamental to a wide array of biological processes, including signal transduction, immune responses, and metabolism, all of which are critical for organismal health. Accurate PPI prediction is crucial for drug discovery, as it aids in identifying potential therapeutic targets (Zinzalla & Thurston, 2009), understanding disease mechanisms by revealing the molecular basis of pathologies (Kuzmanov & Emili, 2013), and enhancing synthetic biology by enabling the design of new drugs (J. Bienstock, 2012; Li et al., 2014).

Early learning-based PPI prediction models (Chen & Liu, 2005; Lin & Chen, 2013; Sarkar & Saha, 2019) rely on hand-crafted protein features, such as amino acid composition or physicochemical properties. However, they suffer from limited PPI representation quality and struggle to scale to large PPI datasets. Recent advancements leverage deep learning to automatically extract protein representations from a specific protein modality, such as sequence (Wang et al., 2019; Zhang et al., 2019a), structure (Singh et al., 2010; Zhang et al., 2012), and function (Zhang & Tang, 2016; Wu et al., 2006). Furthermore, graph neural networks (GNNs) (Wu et al., 2020) are used to capture the relational dependencies in PPI networks (Gao et al., 2023; Wu et al., 2024; Tang et al., 2025; Chen et al., 2025a; Yang et al., 2025).

Despite these recent advances, existing methods overlook two key aspects. First, proteins inherently exhibit hierarchical organization: (i) micro-scale residues, (ii) meso-scale functional motifs, and (iii) macro-scale whole protein. Crucially, PPIs are often mediated by specific meso-scale motifs within the protein structures. These motifs represent fine-grained dependencies that regulate the protein's behavior, influencing how proteins recognize and bind to one another (Ciriello & Guerra, 2008; Ivarsson & Jemth, 2019). However, most PPI prediction methods (Wu et al., 2024; Tang et al., 2025; Chen et al., 2025a; Yang et al., 2025) focus on encoding proteins in a flat manner, and obtain protein embeddings by simple averaging of the residue representations. This ignores the crucial meso-scale motif information, and does not capture specific motif patterns that actually drive protein interactions. Though some non-deep learning studies (Bardwell & Treisman, 1994; Ivarsson & Jemth, 2019) explored the role of motifs in PPIs, they rely on hand-crafted motif representations, which limit them to small datasets and fail to account for the distinct functional behaviors of motifs driven by their contextual positioning (Blikstad & Ivarsson, 2015; Seo & Kim, 2018). Thus, there remains a need for a

---

[1]Department of Computer Science and Engineering, The Hong Kong University of Science and Technology, Clear Water Bay, Hong Kong. [2]Lee Shau Kee School of Business and Administration, The Hong Kong Metropolitan University, Ho Man Tin, Kowloon, Hong Kong. Correspondence to: James Kwok <jamesk@cse.ust.hk>.

*Proceedings of the 43$^{rd}$ International Conference on Machine Learning*, Seoul, South Korea. PMLR 306, 2026. Copyright 2026 by the author(s).

hierarchical paradigm to bridge the gap between micro-scale residues and macro-scale protein representations by capturing meso-scale motif semantics and their varying contextual importance for large-scale PPI prediction.

The second aspect often overlooked by existing methods is that each protein modality (e.g., sequence, structure, or function) contains relatively orthogonal information for PPI. The sequence modality provides insights into the primary structure of proteins, capturing amino acid sequences that dictate protein folding and interaction potential. The structural modality reveals the 3D conformation, highlighting spatial arrangements that determine the protein's interaction behavior. The functional modality captures the biochemical properties and cellular roles, which are critical for the mechanisms behind protein interactions. Integrating these complementary semantics can offer a more comprehensive protein representation (Hamamsy et al., 2023; Fan et al., 2025; Hu et al., 2023). However, preliminary studies (Hu et al., 2023; Chen et al., 2025a) use simple concatenation of modality-specific information and only yield some PPI prediction performance improvements. While MASSA (Hu et al., 2023) integrates protein sequence, structure, and functional annotations, it relies on a coarse-grained global fusion strategy that overlooks the critical role of meso-scale motifs. Consequently, the synergistic potential of these modalities is diluted in the flat representation. Thus, how to achieve fine-grained multimodal integration that respects the hierarchical nature of proteins remains an open challenge.

To address these limitations, we present a *Hierarchical Motif-based MultiModal protein Encoder for PPI Prediction* (MMM-PPI). Unlike previous methods that rely on flat, coarse-grained representations, MMM-PPI considers the intrinsic biological hierarchy of proteins with multimodal semantics, constructing embeddings in a bottom-up, multimodal manner. (i) **Micro-scale**: Extract residue features from the sequence, structure, and function modalities by modality-specific encoders; (ii) **Meso-scale**: Introduce a multimodal motif encoder to aggregate residue features into spatially-informed motif embeddings; (iii) **Macro-scale**: Introduce a multimodal protein encoder to integrate motifs via pairwise motif co-attention, capturing both the synergistic effects of multiple PPI modalities and fine-grained motif dependencies that govern interactions. Once the encoder is pre-trained, it is used off-the-shelf to generate protein embeddings, which serve as enriched initial node features for a PPI network and are then encoded by a GNN to predict PPIs. Extensive experiments on three PPI datasets demonstrate that MMM-PPI achieves state-of-the-art performance on multi-label PPI prediction, particularly in scenarios with challenging data partitions and limited training data.

In summary, this paper makes the following contributions:

- We propose a novel multimodal motif encoder that trans-

forms single-modality residue encodings into multimodal motif embeddings.

- We introduce a novel protein encoder with motif co-attention that fuses multimodal motif embeddings, jointly capturing motif importance and inter-modal correlations.

- Extensive experiments on three PPI datasets show that our approach achieves state-of-the-art performance, especially under challenging data partitions and limited training data.

## 2. Related Work

### 2.1. Protein Protein Interaction (PPI) Prediction

Early efforts in PPI prediction (Chen & Liu, 2005; Lin & Chen, 2013; Sarkar & Saha, 2019) predominantly rely on using manually designed features (such as amino acid composition and physicochemical properties) from protein sequences as input for pairwise PPI classifiers to infer potential interactions. However, they are often limited by the quality and comprehensiveness of the handcrafted features, which can miss critical interaction determinants.

As the field evolves, a key highlight is the application of graph neural networks (GNNs) (Wu et al., 2020). GNNs exploit the complex relationships in PPI networks (which treat proteins as nodes and their interactions as edges) and enable more accurate predictions and discovery of previously unknown PPIs (Lv et al., 2021; Zhao et al., 2023). Recent GNN-based methods further extend the initial protein representations from static structure (Gao et al., 2023) by incorporating conformational dynamics (Chen et al., 2025b), hyperbolic geometry (Tang et al., 2025), and microenvironment-aware patterns (Wu et al., 2024; Chen et al., 2025a; Yang et al., 2025) to capture more intricate biological signals.

Despite the innovations, existing methods rely on coarse-grained embeddings derived from a flattened protein encoding process and underutilized multimodal information. This hinders the ability to capture complementary information on multiple modalities and fine-grained motif information.

### 2.2. Protein Representation Learning

Many studies try to learn high-quality protein representations by leveraging various pre-training tasks to capture different aspects of protein information, including sequence (Lin et al., 2023), structure (Zhang et al., 2023), and function (Zhang et al., 2022; Zhou et al., 2023; Xu et al., 2023), or a combination of them (Su et al., 2024; Hu et al., 2023). For instance, MASSA (Hu et al., 2023) integrates them using a two-step alignment method. Although this achieves some performance gain on benchmark tasks such as protein property prediction and PPI prediction, MASSA performs multimodal integration at a coarse-grained protein

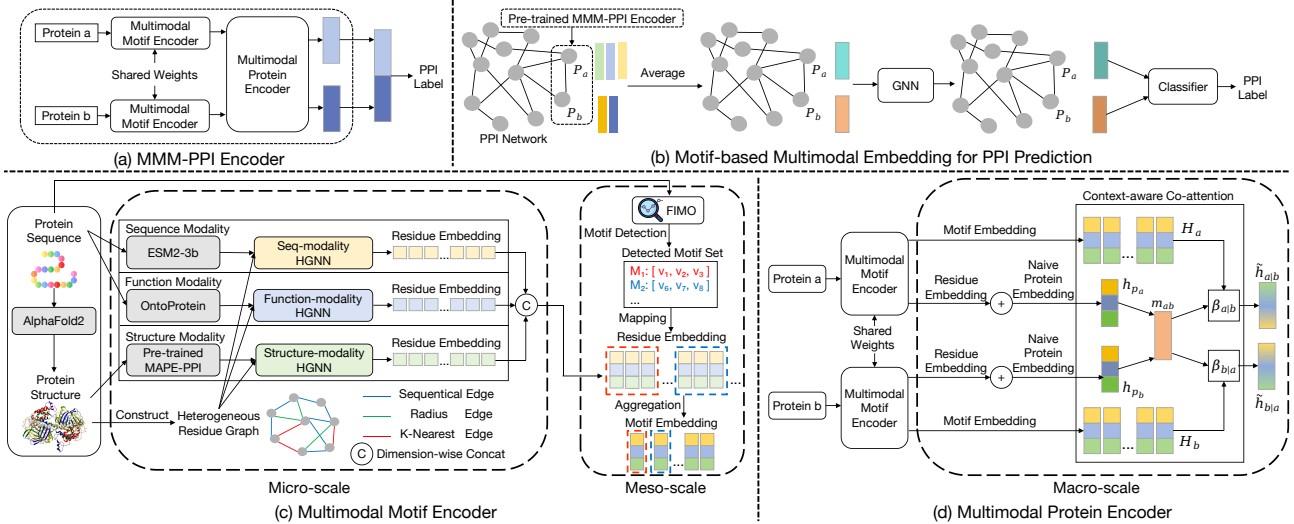

*Figure 1.* Overview of the MMM-PPI. (a) The protein pair is encoded using the hierarchical encoding process based on the multimodal motif encoder and multimodal protein encoder. (b) For PPI prediction, the pre-trained encoder is used to generate embeddings for the proteins, which are processed through a GNN for classification. (c) Overview of the multimodal motif encoder. (d) Overview of the multimodal protein encoder. Different colors denote different modalities (yellow for sequence, blue for function, and green for structure).

level, neglecting the meso-scale motifs where interactions actually happen. This results in the loss of specific local information within the global representation.

## 3. Proposed Method

We are given a set of proteins $\mathcal{P} = \{P_1, \ldots, P_m\}$ and a set of PPI interactions $\mathcal{X} = \{x_{ij} = (P_i, P_j) \mid i \neq j, P_i, P_j \in \mathcal{P}\}$, where each interaction is associated with a label set $y_{ij} \subseteq \mathcal{C}$ and $\mathcal{C}$ is the set of all interaction labels. The dataset $\mathcal{X}$ is divided into a training set $\mathcal{X}_K$ (with known interaction labels $\mathcal{Y}_K$) and test set $\mathcal{X}_U$ (with unknown labels). PPI prediction aims to predict the multi-label interactions between proteins. In this paper, we consider both the transductive and inductive settings. In the transductive setting, the model is trained on $(\mathcal{X}, \mathcal{Y}_K)$ and used to predict labels $\hat{y}_{ij}$ for pairs in $\mathcal{X}_U$. In the inductive setting, model training only uses $(\mathcal{X}_K, \mathcal{Y}_K)$, without access to the full set $\mathcal{X}$.

### 3.1. Multimodal Hierarchical Protein Encoding

PPIs are strongly influenced by the protein's spatial structure, which defines key binding sites and determines the interaction between motifs (Kleywegt, 1999). Moreover, the same motif may exhibit distinct behaviors in different proteins, influenced by factors such as the overall folding pattern, local environment, and interactions with other domains (Dreier et al., 2022; Mason et al., 2010). Thus, motif representations should be adaptive to the specific spatial context of each amino acid residue constituting the motif.

In this section, we propose constructing the protein representations in a **bottom-up multi-scale** manner. In Section 3.1.1, we first encode micro-scale residue features from the

sequence, structure, and function modalities. Section 3.1.2 then aggregates them to generate meso-scale multimodal motif embeddings. Finally, Section 3.1.3 integrates the motif embeddings to a macro-scale protein representation via a context-aware co-attention mechanism. The whole pipeline is shown in Figure 1.

#### 3.1.1. MICRO-SCALE: RESIDUE ENCODING

Since motifs are composed of amino acid residues, we first consider how to learn residue embeddings. Consider a protein $P_i$ with residue sequence $S_i = (v_1, v_2, \ldots, v_{N_i})$, where each $v_j$ is an amino acid residue and $N_i$ is the number of residues in $P_i$. For high efficiency and structural consistency, we directly use the 3D residue coordinates provided by Wu et al. (2024), which were retrieved from the AlphaFold Protein Structure Database (Jumper et al., 2021; Varadi et al., 2022). The protein is then represented as a graph $\mathcal{G}_{P_i} = (\mathcal{V}_{P_i}, \mathcal{E}_{P_i})$, where each node in $\mathcal{V}_{P_i}$ corresponds to a residue of $P_i$ and each edge in $\mathcal{E}_{P_i}$ contains connections between the residues. We adopt the three edge types in Zhang et al. (2023): (i) sequential edges, which link consecutive residues in the residue sequence; (ii) radius edges, which connect residues whose spatial Euclidean distance is below a predefined threshold; and (iii) $K$-nearest edges, which link each residue to its $K$ nearest neighbors.

Current PPI prediction methods (Lv et al., 2021; Gao et al., 2023; Wu et al., 2024; Chen et al., 2025a; Yang et al., 2025) rely on representations from one or two modalities, which may miss their rich and complementary information. In this paper, we propose to integrate the sequence, structure, and function modalities to provide a more comprehensive view.

For each modality, one heterogeneous graph neural network (HGNN) (Zhang et al., 2019b) is used on the shared $\mathcal{G}_{P_i}$ to produce residue embeddings specific to protein $P_i$. The initial embedding $\mathbf{h}_v^{(0)}$ for node $v$ is derived from the corresponding frozen modality-specific pre-trained model. In the experiments, we use (near-) SOTA modality-specific models that have been pre-trained on large-scale unlabeled data: (i) **ESM2-3b** (Lin et al., 2023) for sequence; (ii) **Pre-trained MAPE-PPI** (Wu et al., 2024) for structure; and (iii) **OntoProtein** (Zhang et al., 2022) for function. Note that the proposed method is agnostic to the choice of these pre-trained models.

The embedding of node (i.e., residue) $v$ at the $l$-layer is updated as:

$$\mathbf{h}_v^{(l)}=\mathrm{BN}\left(\mathrm{ReLU}\left(\mathbf{W}_h^{(l)}\cdot\sum_{r\in\mathcal{R}}\mathbf{W}_r^{(l)}\sum_{u\in\mathcal{N}_r(v)}\mathbf{h}_u^{(l-1)}\right)\right),\quad(1)$$

where $\mathcal{N}_r(v)$ is the set of nodes connected to node $v$ via edge type $r$, $\mathbf{W}_r^{(l)}, \mathbf{W}_h^{(l)}$ are weight matrices, and BN is batch normalization. After aggregation, $\mathbf{h}_v^{(L)}$ is used as residue $v$'s embedding for that modality. Finally, we concatenate embeddings from all three modalities to obtain a $D$-dimensional multimodal residual embedding $\mathbf{h}_v$ of $v$. This pipeline is summarized in Figure 1(c).

### 3.1.2. Meso-scale: Motif Construction

Given protein $P_i$, we use the Find Individual Motif Occurrences (FIMO) algorithm (Grant et al., 2011) from the MEME suite (Bailey et al., 2009) to identify its motifs $\{M_{i_1}, M_{i_2}, \ldots, M_{i_k}\}$, where $i_k$ is the number of motifs in $P_i$. FIMO is efficient, and scans each protein only once. More details on motif detection are in Appendix A.

Next, we obtain the embedding of each motif $M_{i_j}$ (with residues $[v_{ij_s}, \ldots, v_{ij_e}]$, where $v_{ij_s}$ and $v_{ij_e}$ are the start and end residues of $M_{i_j}$) as: $\mathbf{h}_{M_{i_j}} = \mathbf{h}_{v_{ij_s}} + \cdots + \mathbf{h}_{v_{ij_e}}$, where $\{\mathbf{h}_{v_{ij_s}}, \ldots, \mathbf{h}_{v_{ij_e}}\}$ are residue embeddings obtained in Section 3.1.1. The whole set of $P_i$'s motif embeddings obtained can be represented by the matrix (Figure 1(c)):

$$\mathbf{H}_i = [\mathbf{h}_{M_{i_1}}, \mathbf{h}_{M_{i_2}}, \ldots, \mathbf{h}_{M_{i_k}}] \in \mathbb{R}^{D\times i_k}.\quad(2)$$

### 3.1.3. Macro-Scale: Protein Encoding

For a protein $P_i$ with residues $\{v_1, v_2, \ldots, v_{N_i}\}$, previous works (Lv et al., 2021; Gao et al., 2023; Wu et al., 2024; Chen et al., 2025a; Yang et al., 2025) encode $P_i$ by simply aggregating its constituent residue embeddings:

$$\mathbf{h}_{P_i} = \mathbf{h}_{v_1} + \mathbf{h}_{v_2} + \cdots + \mathbf{h}_{v_{N_i}}.\quad(3)$$

With the motif embeddings in (2), we can also analogously obtain the protein embedding by similarly aggregating the motif embeddings. However, different motifs have different importance in PPI prediction. They also exhibit distinct functional behaviors depending on specific interacting proteins and contextual positionings within different proteins (Blikstad & Ivarsson, 2015; Seo & Kim, 2018). Thus, to achieve a more informative protein embedding, protein-specific weights should be associated with the motifs.

Inspired by the success of question-image co-attention in VQA (Lu et al., 2016; Yu et al., 2019), we design a novel pairwise context-aware motif co-attention mechanism to calculate the protein-specific weights for the motifs (Figure 1(d)). Unlike typical VQA co-attention (Lu et al., 2016; Yu et al., 2019) that rely on dense, computation-intensive alignment to model pairwise similarities between every word and visual region, our method avoids direct motif-to-motif alignment. Instead, for each protein pair $(P_a, P_b)$, we use a basic interaction vector $\mathbf{m}_{ab}$ as a global condition to adjust the motif weights. These motif weights represent how much each motif contributes to the overall interaction, rather than how well it aligns with specific motifs in the partner protein. Based on this mechanism, we generate partner-specific feature vectors $\tilde{\mathbf{h}}_{a|b}$ and $\tilde{\mathbf{h}}_{b|a}$. This captures the bidirectional dependency, where the motif weights in $P_a$ are conditioned on the context of $P_b$, and vice versa.

Specifically, let $\mathbf{h}_{P_a}$ and $\mathbf{h}_{P_b}$ be the two proteins' naive embeddings in (3). We first obtain their basic interaction $\mathbf{m}_{ab} = \mathbf{h}_{P_a} \odot \mathbf{h}_{P_b}$, where $\odot$ is the element-wise product. The representation of $P_a$ (with motif embedding matrix $\mathbf{H}_a$) is a weighted sum of its constituent motif embeddings:

$$\tilde{\mathbf{h}}_{a|b} = \tanh(\mathbf{H}_a \boldsymbol{\beta}_{a|b}^\top),\quad(4)$$

where $\boldsymbol{\beta}_{a|b}$ contains the attention weights of motifs in $P_a$ to $\mathbf{m}_{ab}$ in a $U$-dimensional latent space:

$$\begin{aligned}\boldsymbol{\beta}_{a|b} &= \mathrm{softmax}(\mathbf{w}_a^\top \mathbf{h}_{a|b}), \\ \mathbf{h}_{a|b} &= \tanh(\mathbf{W}_a^q \mathbf{H}_a) \odot \tanh(\mathbf{W}_a^m \mathbf{m}_{ab}),\end{aligned}$$

where $\mathbf{w}_a \in \mathbb{R}^U$, and $\mathbf{W}_a^q, \mathbf{W}_a^m \in \mathbb{R}^{U\times D}$. For $P_b$, one can obtain $\tilde{\mathbf{h}}_{b|a}$ analogously.

Recall that the motif embeddings in Section 3.1.2 are derived from residue representations which are already spatially informed by the upstream 3D-aware HGNNs. Since the local geometric context is encoded within the residues' features, explicitly incorporating additional contact-level geometric during motif aggregation becomes computationally redundant. Thus, the co-attention mechanism here focuses on modeling the semantic relevance and cross-modal correlations, rather than re-encoding structural dependencies.

### 3.2. Model Learning

We first construct the PPI graph $\mathcal{G} = (\mathcal{P}, \mathcal{X})$ (resp. $\mathcal{G} = (\mathcal{P}, \mathcal{X}_\mathcal{K})$) for the transductive (resp. inductive) setting,

where each node is a protein and each edge is an PPI interaction. The protein representations produced by the encoder in Section 3.1.3 are used as initial node features for $\mathcal{G}$. Since the representation of protein $P_a$ depends on its interacting proteins, we obtain its representation as

$$\mathbf{h}_{P_a} = \frac{1}{|\mathcal{N}(P_a)|} \sum_{P_k \in \mathcal{N}(P_a)} \tilde{\mathbf{h}}_{a|k},$$

where $\mathcal{N}(P_a)$ is the set of proteins interacting with $P_a$. A GNN is applied on the PPI graph, with a linear layer used to predict the PPI labels. In the experiments, we use the Graph Isomorphism Network (Xu et al., 2019).

Following MAPE-PPI (Wu et al., 2024), we decouple model training into two stages. The protein encoder is first pre-trained. It is then frozen and the GNN is trained. This avoid the high cost of end-to-end training, which will redundantly re-compute the embeddings for every interactions at every GNN training step. Our two-stage paradigm compute embeddings once as frozen features, reducing the complexity from a multiplicative dependency (encoding × GNN steps) to an additive one, ensuring high efficiency and scalability.

To pre-train the protein encoder, for each protein pair $(P_a, P_b)$ in the training set $\mathcal{X}_K$, we concatenate $\tilde{\mathbf{h}}_{a|b}$ and $\tilde{\mathbf{h}}_{b|a}$ obtained from (4) and feed it to a fully-connected linear layer (FCL) for PPI prediction:

$$\hat{\mathbf{y}}_{ab} = \mathrm{FCL}(\mathrm{concat}[\tilde{\mathbf{h}}_{a|b}, \tilde{\mathbf{h}}_{b|a}]).$$

Parameters in both the multimodal motif encoder (the three HGNNs) and multimodal protein encoder (matrices $\mathbf{W}_a^q$, $\mathbf{W}_a^m$, $\mathbf{w}_a$, $\mathbf{W}_b^q$, $\mathbf{W}_b^m$, $\mathbf{w}_b$) are then learned by minimizing the cross-entropy loss. More details are in Appendix B. The pseudo-code is shown in Algorithm 1 of Appendix C.

# 4. Experiments

## 4.1. Setup

**Datasets.** Experiments are performed on three PPI datasets that have been commonly used (Gao et al., 2023; Wu et al., 2024; Chen et al., 2025a; Yang et al., 2025): STRING, SHS27k, and SHS148k (Table 1). The STRING dataset is derived from the STRING database (Szklarczyk et al., 2019) and is the **largest** publicly available PPI dataset to our knowledge. SHS27k and SHS148k (Chen et al., 2019) are more challenging datasets, generated by randomly selecting proteins with more than 50 amino acids and less than 40% sequence identity from the Homo sapiens subset of STRING.

As in (Wu et al., 2024), these datasets are split into the training, validation, and test sets in the ratio 6:2:2 using three

*Table 1.* Statistics of PPI datasets.

| Dataset | #Proteins | #Interactions | #PPI Entries |
|---------|-----------|---------------|--------------|
| STRING | 14,952 | 572,568 | 1,150,830 |
| SHS27k | 1,663 | 7,401 | 16,912 |
| SHS148k | 5,082 | 43,397 | 99,782 |

partitioning strategies (Lv et al., 2021): Random, Breadth-First Search (BFS), and Depth-First Search (DFS). Random splitting assigns data stochastically; BFS simulates densely interconnected unknown protein clusters, while DFS targets sparsely distributed unknown protein clusters. More details are in Appendix D.

We further statistic the counts and lengths of the extracted motifsin Appendix A. The analysis shows that the distributions of motif counts (Table 10) and lengths (Table 11) are consistent across the three datasets with different scales.

**Baselines.** Recent PPI prediction baselines can be divided into two categories based on whether the training process includes additional pre-training. Standard baselines, such as DPPI (Hashemifar et al., 2018), DNN-PPI (Li et al., 2018), PIPR (Chen et al., 2019), GNN-PPI (Lv et al., 2021), SemiGNN-PPI (Zhao et al., 2023), and HIGH-PPI (Gao et al., 2023), have only one stage and do not include additional pre-training. The second set of baselines requires additional pre-training stages or pre-training data before PPI prediction. They include (i) single-modality pre-trained models: DCMF-PPI (Chen et al., 2025b), ESM2-3b (Lin et al., 2023), pre-trained MAPE-PPI (Wu et al., 2024), and HI-PPI (Tang et al., 2025); (ii) multi-modality pre-trained models: OntoProtein (Zhang et al., 2022), KeAP (Zhou et al., 2023), ProtST (Xu et al., 2023), MASSA (Hu et al., 2023), SaProt (Su et al., 2024), MEGAE (Chen et al., 2025a), and MicroEnvPPI (Yang et al., 2025).

Since the proposed model has to pre-train the protein encoder, it is more similar to the second set of baselines, which typically also outperforms the first set. Thus, here we focus on the second set of of baselines. Results on the first set of baselines are in Appendix E. Besides, as shown in DPPI (Hashemifar et al., 2018), DPPI outperforms traditional non-deep learning baselines. As will be shown in Table 13, the proposed MMM-PPI outperforms DPPI. Hence, we do not compare with the non-deep methods here.

**Implementation Details.** In the GNN-based PPI prediction literature (Lv et al., 2021; Zhao et al., 2023; Gao et al., 2023; Wu et al., 2024), usually only the transductive setting is reported. However, in practical PPI applications, one may face new protein pairs that have never appeared in the training set. Hence, we also experiment with the inductive setting, which trains the GNN on the graph constructed from the training data and performs testing on unseen edges.

The experiment is repeated 3 times with different random seeds. For each run, following MAPE-PPI (Wu et al., 2024), we select the model with the best validation performance for testing. For performance evaluation, we use the Micro-F1 score as in (Lv et al., 2021; Zhao et al., 2023; Gao et al., 2023; Wu et al., 2024), and report the average over the 3 runs. Due to the lack of space, additional evaluation us-

*Table 2.* Micro-F1 scores (%) of various methods on different datasets and partitions with the transductive and inductive setting. **Bold** and underline represent the top and second-best performance metrics. SEQ: sequence, STR: structure, FUNC: function.

| Setting | Method | Modality | | | SHS27k | | | SHS148k | | | STRING | | | Avg. |
|---|---|---|---|---|---|---|---|---|---|---|---|---|---|---|
| | | SEQ | STR | FUNC | Random | DFS | BFS | Random | DFS | BFS | Random | DFS | BFS | |
| transductive | DCMF-PPI | ✓ | | | 84.15 | 69.25 | 70.84 | 91.88 | 76.89 | 74.84 | 95.12 | 84.77 | 78.57 | 80.70 |
| | ESM2-3b | ✓ | | | 88.19 | 71.47 | 74.08 | 92.63 | 81.16 | 75.08 | 96.20 | 88.72 | 79.56 | 83.01 |
| | MAPE-PPI | | ✓ | | 89.04 | 73.93 | 74.23 | 92.86 | 82.15 | 76.26 | 96.60 | 88.71 | 80.27 | 83.78 |
| | HI-PPI | | ✓ | | 87.84 | 74.05 | 74.53 | 92.75 | 81.77 | 76.27 | 96.53 | 89.03 | 81.45 | 83.80 |
| | OntoProtein | ✓ | | ✓ | 88.42 | 72.10 | 74.93 | 92.67 | 82.03 | 75.75 | 96.55 | 88.16 | 80.32 | 83.44 |
| | KeAP | ✓ | | ✓ | 88.62 | 72.43 | 75.82 | 92.73 | 83.04 | 75.59 | 96.04 | 88.87 | 81.49 | 83.85 |
| | ProtST | ✓ | | ✓ | 88.54 | 73.66 | 74.98 | 92.07 | 82.33 | 76.17 | 96.22 | 88.79 | 81.10 | 83.76 |
| | SaProt | ✓ | ✓ | | 88.22 | 73.72 | 75.50 | 92.66 | 82.94 | 76.29 | 96.46 | 89.36 | 82.04 | 84.13 |
| | MEGAE | ✓ | ✓ | | 88.08 | 72.77 | 73.71 | 92.26 | 83.02 | 75.53 | 96.40 | 88.95 | 81.34 | 83.56 |
| | MicroEnvPPI | ✓ | ✓ | | 88.28 | 74.57 | 76.22 | 92.89 | 83.21 | 76.56 | 96.60 | 89.13 | 81.20 | 84.30 |
| | MASSA | ✓ | ✓ | ✓ | 88.43 | 73.75 | 76.67 | 91.04 | 83.10 | 76.31 | 96.07 | 89.47 | 81.54 | 84.04 |
| | MMM-PPI | ✓ | ✓ | ✓ | **89.25** | **75.93** | **79.09** | **93.38** | **84.24** | **77.70** | **97.05** | **90.62** | **83.18** | **85.60** |
| inductive | DCMF-PPI | ✓ | | | 82.14 | 68.80 | 71.13 | 83.58 | 75.48 | 61.22 | 89.44 | 81.20 | 71.42 | 76.05 |
| | ESM2-3b | ✓ | | | 83.05 | 69.81 | 73.16 | 86.66 | 78.04 | 65.08 | 92.02 | 83.94 | 77.13 | 78.77 |
| | MAPE-PPI | | ✓ | | 82.60 | 70.14 | 72.03 | 84.35 | 73.61 | 62.01 | 90.40 | 82.55 | 71.25 | 76.55 |
| | HI-PPI | | ✓ | | 83.15 | 70.29 | 72.65 | 84.89 | 76.01 | 63.88 | 91.28 | 82.87 | 72.06 | 77.45 |
| | OntoProtein | ✓ | | ✓ | 83.63 | 69.73 | 72.04 | 87.73 | 77.92 | 64.54 | 91.46 | 84.70 | 76.44 | 78.69 |
| | KeAP | ✓ | | ✓ | 84.79 | 70.16 | 73.25 | 88.98 | 78.12 | 65.80 | 93.18 | 85.91 | **77.26** | 79.72 |
| | ProtST | ✓ | | ✓ | 83.81 | 70.45 | 73.19 | 88.66 | 79.65 | 64.14 | 92.78 | 84.05 | 75.39 | 79.12 |
| | SaProt | ✓ | ✓ | | 84.12 | 70.81 | 73.29 | 87.75 | 79.83 | 65.20 | 92.46 | 84.39 | 75.03 | 79.21 |
| | MEGAE | ✓ | ✓ | | 83.51 | 69.56 | 71.89 | 84.52 | 78.89 | 62.98 | 93.30 | 82.72 | 72.57 | 77.77 |
| | MicroEnvPPI | ✓ | ✓ | | 83.83 | 70.71 | 73.25 | 88.29 | 79.81 | 64.36 | 93.75 | 84.34 | 75.09 | 79.27 |
| | MASSA | ✓ | ✓ | ✓ | 83.74 | 71.69 | 74.19 | 87.95 | 80.83 | 65.77 | 92.06 | 86.60 | 77.08 | 79.99 |
| | MMM-PPI | ✓ | ✓ | ✓ | **87.27** | **73.82** | **75.63** | **92.04** | **82.33** | **67.34** | **95.69** | **87.87** | 76.53 | **82.06** |

ing the AUC and AUPR metrics are shown in Appendix F. More implementation details are in Appendix G, and qualitative analysis on biological interpretations between the functional roles of high-attention motifs and nature of the corresponding PPIs is in Appendix H.

## 4.2. PPI Prediction Performance

We first evaluate the model in the transductive setting. Table 2 shows the micro-F1 scores. As can be seen, the proposed MMM-PPI achieves the highest scores on all datasets, particularly under the more difficult BFS and DFS splits. Notably, MMM-PPI significantly outperforms not only single-modality pre-trained models but also multi-modality ones including MASSA, which incorporates three modalities but does so at a coarse-grained global level. This illustrates the limitation of flat integration and highlights the importance of using meso-scale structure in our hierarchical encoding.

## 4.3. Generalizing to Unseen PPIs

While the transductive setting evaluates performance on a fixed graph, practical applications often require predicting interactions that are not present during training. To assess this capability, we further evaluate the model in the inductive setting. The micro-F1 results are shown in Table 2. As can be seen, MMM-PPI continues to achieve a significant per-

formance advantage over the baselines, reflecting its robust generalization ability to unseen PPIs. Besides, while many baselines (such as KeAP and MASSA) show significant fluctuations in their performance depending on the partitioning strategy, MMM-PPI remains the best-performing method in nearly all cases. This demonstrates that its effectiveness is not sensitive to the partitioning method.

## 4.4. Generalizing to Unseen Proteins

Besides generalizing to unseen interactions, a bigger challenge is generalizing to entirely unseen proteins. In this experiment, following (Lv et al., 2021; Zhao et al., 2023; Wu et al., 2024), we divide the test data of SHS27k into three non-overlapping subsets based on whether the two interacting proteins have appeared in the training data or not: (i) BS (both seen): Both proteins have appeared in the training data; (ii) ES (either seen): Only one protein has appeared; (iii) NS (neither seen): Neither protein has appeared in the training data. Note that the BFS and DFS partition schemes are more difficult than random partitioning because they only contain the ES and NS subsets (Lv et al., 2021).

We select the five strongest single-modality and multi-modality baselines in Table 2 for comparison: (i) ESM2, which encodes sequence information; (ii) MAPE-PPI, which encodes structural information; (iii) KeAP, which encodes

*Table 3.* Micro-F1 score (%) performance of different models on different edge subsets (BS, ES, and NS) on SHS27k dataset. The ratio in parentheses indicates the proportion of this subset in the test set. The training setting is transductive.

| Method | label ratio | Random | | | | DFS | | | BFS | | |
|---|---|---|---|---|---|---|---|---|---|---|---|
| | | BS (89.94%) | ES (9.66%) | NS (0.41%) | Avg. | ES (76.57%) | NS (23.43%) | Avg. | ES (66.80%) | NS (33.20%) | Avg. |
| ESM2-3b | 100% | 89.43 | 68.92 | 50.00 | 88.16 | 74.22 | 63.71 | 72.09 | 75.54 | 70.53 | 74.12 |
| MAPE-PPI | | 90.32 | 70.75 | 40.00 | 88.99 | 75.89 | 64.57 | 73.53 | 78.06 | 66.79 | 74.12 |
| KeAP | | 90.09 | 70.29 | 55.56 | 88.57 | 74.07 | 66.76 | 72.48 | 76.92 | 73.98 | 75.86 |
| MicroEnvPPI | | 89.61 | 70.75 | 57.14 | 88.25 | 76.05 | 67.80 | 74.26 | 77.08 | 75.04 | 76.40 |
| MASSA | | 89.33 | 70.70 | 60.00 | 88.28 | 75.57 | 67.37 | 73.93 | 77.87 | 74.51 | 76.56 |
| MMM-PPI | | **90.62** | **71.30** | **66.67** | **89.20** | **77.69** | **68.16** | **75.70** | **79.95** | **78.52** | **79.48** |
| | | BS (57.19%) | ES (37.54%) | NS (5.27%) | Avg. | ES (56.15%) | NS (43.84%) | Avg. | ES (53.01%) | NS (46.99%) | Avg. |
| ESM2-3b | 20% | 80.09 | 71.74 | 50.99 | 76.18 | 64.78 | 64.96 | 64.86 | 68.13 | 67.99 | 68.07 |
| MAPE-PPI | | 82.47 | 71.52 | 50.99 | 78.86 | 69.62 | 64.15 | 67.43 | 67.71 | **72.16** | 69.77 |
| KeAP | | 83.76 | 71.03 | 48.33 | 77.78 | 69.94 | 64.98 | 67.90 | 69.06 | 69.90 | 69.51 |
| MicroEnvPPI | | 82.76 | 72.55 | 47.46 | 78.31 | 70.98 | 64.61 | 68.42 | 69.98 | 69.85 | 69.92 |
| MASSA | | 82.13 | 72.08 | 49.06 | 77.92 | 68.95 | 64.38 | 67.22 | 69.35 | 69.50 | 69.42 |
| MMM-PPI | | **85.84** | **73.02** | **52.17** | **80.14** | **72.82** | **65.79** | **70.07** | **70.61** | 71.22 | **70.88** |
| | | BS (22.69%) | ES (50.91%) | NS (26.40%) | Avg. | ES (33.47%) | NS (66.53%) | Avg. | ES (32.06%) | NS (67.93%) | Avg. |
| ESM2-3b | 5% | 73.18 | 67.33 | 53.60 | 66.00 | 58.63 | 65.82 | 63.27 | 60.77 | 65.25 | 63.76 |
| MAPE-PPI | | 69.81 | 67.17 | 52.44 | 64.69 | 62.70 | 63.85 | 63.42 | 59.94 | 66.64 | 64.61 |
| KeAP | | 74.17 | 67.37 | 55.22 | 65.82 | 61.64 | 65.36 | 64.18 | 63.61 | 66.88 | 65.94 |
| MicroEnvPPI | | 75.54 | 68.91 | 53.33 | 67.46 | 61.39 | 65.22 | 63.89 | 62.89 | 67.16 | 65.72 |
| MASSA | | 74.44 | 68.53 | 52.37 | 66.97 | 61.46 | 66.16 | 64.51 | 61.46 | 65.36 | 64.12 |
| MMM-PPI | | **77.51** | **69.46** | **56.56** | **68.79** | **62.70** | **67.43** | **65.73** | **64.99** | **68.99** | **67.62** |

sequence and function information; (iv) MicroEnvPPI, which encodes sequence and structure information; (v) MASSA, which encodes sequence, structure, and function information. Moreover, as Table 2 shows similar conclusions in both the transductive and inductive settings, we will only report results on the transductive setting here.

The testing micro-F1 results are shown in Table 3. As can be seen, compared to the baselines, MMM-PPI can better generalize for the ES and NS edges. We attribute this generalization capability to the compositional nature of proteins and the explicit motif modeling in MMM-PPI.

Although the testing proteins do not appear during training, their constituent meso-scale motifs often do. As shown in Table 12 in Appendix A, each unseen protein contains at least one motif that has appeared in the seen proteins. This motif recurrence serves as a strong inductive bias. Unlike traditional models that struggle to generalize from global protein embeddings, our model decomposes novel proteins into familiar local substructures, allowing it to infer interactions based on learned local motif behaviors.

### 4.5. Effect of Labeled Data Availability

As in (Zhao et al., 2023; Wu et al., 2024), we vary the ratio of training labels in $\{5\%, 20\%, 100\%\}$. For the known labels $\mathcal{Y}_K$ of training set $\mathcal{X}_K$, we randomly mask the label set $y_{ij} \in \mathcal{Y}_K$ (corresponding to $x_{ij} \in \mathcal{X}_K$) until the proportion of remaining labels in $\mathcal{Y}_K$ reaches the predefined ratio. As can be seen from Table 3, MMM-PPI outperforms all base-

lines even when data are scarce (e.g., 5%). Again, this is because the shared motifs between seen and unseen proteins can serve as a bridge for the model to predict interactions.

### 4.6. Ablation Studies

In this section, we perform ablation studies to evaluate the usefulness of multimodal information and hierarchical encoding in MMM-PPI. Due to the lack of space, more ablation experiments are in Appendix E.

#### 4.6.1. USEFULNESS OF MULTIMODAL INFORMATION

We compare variants of the proposed model with only one modality for motif/protein encoding. Table 4 shows the testing micro-F1 scores. By comparing rows 2,4,6 with row 10, using three modalities is consistently better than using only one, highlighting the importance of integrating multi-modal protein representations. More details about the complementarity of the three modalities are in Appendix J.

#### 4.6.2. USEFULNESS OF HIERARCHICAL ENCODING

In this experiment, we disable the multimodal motif and/or protein encoder. When both encoders are not used, the micro-scale residue embeddings are directly aggregated into global protein representations using equation (3). When only the motif encoder is disabled, the motif embedding used by the protein encoder is computed by directly averaging its residue embeddings. When only the protein encoder is disabled, the motif embeddings in equation (2) are uni-

*Table 4.* Ablation study (Micro-F1 score, %) on the contributions of multi-modal information (SEQ: sequence, STR: structure, FUNC: function), and the multimodal motif/protein encoders. The transductive setting is used.

| Row Index | Encoder | | Modality | | | SHS27k | | | SHS148k | | | STRING | | | Avg. |
|---|---|---|---|---|---|---|---|---|---|---|---|---|---|---|---|
| | motif | protein | SEQ | STR | FUNC | Random | DFS | BFS | Random | DFS | BFS | Random | DFS | BFS | |
| 1 | ✗ | ✗ | ✓ | ✗ | ✗ | 88.19 | 71.47 | 74.08 | 92.63 | 81.16 | 75.08 | 96.20 | 88.72 | 79.56 | 83.01 |
| 2 | ✓ | ✓ | ✓ | ✗ | ✗ | 88.87 | 73.44 | 78.04 | 92.97 | 82.87 | 76.39 | 96.99 | 90.28 | 82.20 | 84.67 |
| 3 | ✗ | ✗ | ✗ | ✓ | ✗ | 89.04 | 73.93 | 74.23 | 92.86 | 82.15 | 76.26 | 96.60 | 88.71 | 80.27 | 83.78 |
| 4 | ✓ | ✓ | ✗ | ✓ | ✗ | 89.13 | 75.05 | 76.63 | 93.02 | 83.42 | 76.71 | 96.91 | 90.14 | 82.75 | 84.86 |
| 5 | ✗ | ✗ | ✗ | ✗ | ✓ | 88.42 | 72.10 | 74.93 | 92.67 | 82.03 | 75.75 | 96.55 | 88.16 | 80.32 | 83.44 |
| 6 | ✓ | ✓ | ✗ | ✗ | ✓ | 88.96 | 73.95 | 76.62 | 92.80 | 83.58 | 76.79 | 97.01 | 90.38 | 82.38 | 84.72 |
| 7 | ✗ | ✗ | ✓ | ✓ | ✓ | 88.38 | 73.98 | 75.86 | 92.84 | 82.98 | 75.73 | 96.06 | 89.59 | 80.68 | 84.01 |
| 8 | ✓ | ✗ | ✓ | ✓ | ✓ | 88.56 | 74.15 | 76.44 | 92.68 | 83.17 | 76.53 | 95.93 | 89.98 | 81.92 | 84.37 |
| 9 | ✗ | ✓ | ✓ | ✓ | ✓ | 88.96 | 74.73 | 77.53 | 92.90 | 83.88 | 76.64 | 96.58 | 88.31 | 82.26 | 84.64 |
| 10 | ✓ | ✓ | ✓ | ✓ | ✓ | **89.25** | **75.93** | **79.09** | **93.38** | **84.24** | **77.70** | **97.04** | **90.62** | **83.18** | **85.60** |

formly weighted to obtain the protein embedding.

Table 4 shows the testing micro-F1 results in the transductive setting.[1] (i) By comparing rows 1,2, rows 3,4, and rows 5,6, it can be seen that hierarchical encoding improves the performance when a single modality is used. (ii) By comparing rows 7-9, we observe a consistent performance gain when either the motif encoder or the protein encoder is used. This demonstrates that treating proteins as flat residue collections leads to a loss of local semantics, whereas our meso-scale encoding effectively preserves these fine-grained structural dependencies. Finally, the full model (row 10) achieves the best results, showing that the hierarchical construction bridges the gap between local features and global interaction effectively.

### 4.7. Computational Efficiency

In this experiment, we analyze the computational efficiency of our method. We first show the preprocessing cost of FIMO algorithm, and then compare the training time and the memory consumption during the training of MMM-PPI and the recent MAPE-PPI on SHS27k.

The MMM-PPI framework requires motif detection (Section 3.1.2) using the FIMO algorithm. This step is only performed once during preprocessing. As a CPU-based matching

*Table 5.* Running time (in MM:SS) of FIMO algorithm.

| Dataset | Time |
|---|---|
| SHS27k | 03:10 |
| SHS148k | 09:05 |
| STRING | 26:18 |

process, it is highly efficient and does not incur GPU memory costs. Table 5 shows its running time. For example, on the SHS27k dataset, FIMO takes only 3 minutes. In comparison, the pre-training of MMM-PPI takes nearly one hour (as shown in Table 6), indicating that the motif detection cost is very small and almost negligible.

Next, we evaluate the training costs. Both MMM-PPI and

---

[1]Results on the inductive setting are in Appendix E.

*Table 6.* Training time (in MM:SS) and memory consumption (in MiB) of MMM-PPI and MAPE-PPI.

| Method | Encoder Pre-training | | GNN Training | |
|---|---|---|---|---|
| | time | memory | time | memory |
| MAPE-PPI | 40:05 | 15000 | 00:56 | 1800 |
| MMM-PPI (structure) | 49:12 | 14000 | 00:57 | 1600 |
| MMM-PPI (full) | 59:58 | 38000 | 00:56 | 1800 |

MAPE-PPI have two training phases: (i) pre-training the MMM-PPI encoder, and (ii) GNN training. Since MMM-PPI uses three modalities while MAPE-PPI uses only one (structure), we also include a single-modality variant of MMM-PPI with only structure information.

As can be seen in Table 6, the single-modal MMM-PPI variant has similar time and memory usage as MAPE-PPI, confirming that their computational efficiencies are comparable under equivalent input conditions. When extending to the full MMM-PPI with three modalities, the increase in resource demand during encoder pre-training remains modest. The cost increase is a reasonable trade-off for the improved representational richness gained through the integration of three modalities. For GNN training, all three configurations exhibit nearly identical demands on time and memory.

### 4.8. Robustness to Training-Test Structural Similarity

In this experiment, we test whether MMM-PPI's performance gain over the baseline is simply driven by similar protein backbones shared between the training and test sets (Bushuiev et al., 2024). Following (Park & Marcotte, 2012), for each test pair $(P_a, P_b)$ in the test set $\mathcal{X}_U$, we calculate its similarity against every training pair $(P_c, P_d)$ in the training set $\mathcal{X}_K$ and take the maximum score:

$$\text{TM}_{\text{pair}}(P_a, P_b) = \max_{(P_c, P_d) \in \mathcal{X}_K} \Big\{ \min(\text{TM}_{P_a, P_c}, \text{TM}_{P_b, P_d}),$$
$$\min(\text{TM}_{P_a, P_d}, \text{TM}_{P_b, P_c}) \Big\}.$$

Following (Xu & Zhang, 2010), we split the SHS27k test sets on BFS and DFS splitting into *High-similarity* with $\text{TM}_{\text{pair}} \geq 0.5$ and *Low-similarity* with $\text{TM}_{\text{pair}} < 0.5$.

*Table 7.* Micro-F1 (%) on SHS27k test pairs grouped by structural similarity to the training set. High-similarity indicates $\text{TM}_{\text{pair}} \geq 0.5$; and low-similarity indicates $\text{TM}_{\text{pair}} < 0.5$.

| Split | Subset | MicroEnvPPI | MMM-PPI | Δ |
|---|---|---|---|---|
| BFS | High-similarity | 80.12 | 82.05 | +1.93 |
| | Low-similarity | 72.03 | 76.61 | +**4.58** |
| DFS | High-similarity | 76.19 | 77.93 | +1.74 |
| | Low-similarity | 72.88 | 76.20 | +**3.32** |

As shown in Table 7, the performance gap between MMM-PPI and the baseline MicroEnvPPI grows larger on the low-similarity group. This demonstrates that MMM-PPI does not rely on global structural memorization. It generalizes well to unseen protein backbones.

### 4.9. Robustness of Motif Detection

Since MMM-PPI uses a tool algorithm to detect motifs, we check if the model is sensitive to motif detection quality. Motif detection tools can make errors: they can miss true motifs (omission) or report false ones (noise). Therefore, we evaluate the robustness of MMM-PPI by simulating these detection errors.

We perturb the motifs detected by FIMO before passing them to the model. Specifically, we randomly delete a fraction (10%, 20%, or 30%) of the motifs to simulate omission, and randomly add a fraction (10%, 20%, or 30%) of random motifs to simulate noise. We retrain and test MMM-PPI on the SHS27k dataset under these conditions.

*Table 8.* Robustness (Micro-F1, %) of MMM-PPI under random motif omission and noise on SHS27k (transductive setting).

| Variant | Random | DFS | BFS |
|---|---|---|---|
| MMM-PPI (no perturbation) | 89.25 | 75.93 | 79.09 |
| omission 10% | 89.03 | 74.50 | 78.76 |
| omission 20% | 88.48 | 73.17 | 75.98 |
| omission 30% | 88.33 | 73.02 | 75.14 |
| noise 10% | 89.23 | 75.23 | 78.84 |
| noise 20% | 89.04 | 73.80 | 78.31 |
| noise 30% | 88.34 | 73.60 | 78.24 |

Table 8 shows that the performance of MMM-PPI drops only slightly. Even with severe 30% errors, MMM-PPI still outperforms several baselines, e.g., ESM2-3b and MEGAE, in Table 2. This shows that our hierarchical design learns robust representations.

## 5. Biological Interpretability

To evaluate whether MMM-PPI learns meaningful biological features, we perform both qualitative and quantitative analyses. Below, we summarize our qualitative findings and provide a dataset-level quantitative evaluation.

### 5.1. Qualitative Analysis

We examine specific protein pairs to see if the high-attention motifs align with known molecular mechanisms. As de-tailed in Appendix H, the motifs highlighted by MMM-PPI closely match the functional domains driven by the interaction types. This alignment confirms that the model captures interpretable, biologically grounded features.

### 5.2. Quantitative Analysis

Following the protocol in (Nayar et al., 2025), we measure how often the high-attention motifs highlighted by MMM-PPI overlap with functional regions annotated in UniProt (such as active sites or binding sites). A motif is counted as a *Hit* only if it covers at least 50% of the UniProt-annotated region.

For each protein, we calculate the Hit rate for the top-2 highest attention motifs and the bottom-2 lowest attention motifs. For reference, we also add a random baseline that randomly assigns attention weights to the motifs within a protein. This experiment is conducted on the random split of the SHS27k dataset.

*Table 9.* Hit rate (%) between motifs and UniProt-annotated functional regions on SHS27k (50% coverage threshold).

| Method | Top-2 highest | Bottom-2 lowest |
|---|---|---|
| Random attention | 21.58 | 21.60 |
| MMM-PPI | **41.89** | 13.40 |

As shown in the results in Table 9, high-attention motifs from MMM-PPI achieve a Hit rate of 41.89%, which is nearly double the random baseline (21.58%). Meanwhile, low-attention motifs have a Hit rate of only 13.40%, well below the baseline. This demonstrates that MMM-PPI correctly assigns high attention to biologically important regions and low attention to irrelevant ones.

## 6. Conclusion

In this work, we propose MMM-PPI, a novel hierarchical motif-based multimodal protein encoder for enhanced PPI prediction. It addresses two fundamental limitations of existing methods:(i) reliance on flat modeling paradigm that neglects the intrinsic multi-scale organization of proteins, leading to a loss of fine-grained structural semantics; (ii) under-utilization of complementary multimodal information. Our approach builds a bottom-up encoding approach that progressively encodes information from micro-scale multimodal residues to meso-scale functional motifs, and integrates them into macro-scale global embeddings via context-aware co-attention. This design enables the protein embeddings to capture the impact of motifs and the synergistic effect of multiple modalities, facilitating more accurate PPI prediction. Experiments on multiple benchmark PPI datasets demonstrate that the proposed method consistently outperforms state-of-the-art models, particularly under challenging data partitions and low-resource training scenarios.

# Acknowledgment

This research was supported in part by the Research Grants Council of the Hong Kong Special Administrative Region (Grants 16202523 and HKU C7004-22G).

# Impact Statement

This work advances the field of applying machine learning to computational biology by introducing a hierarchical, motif-based, multimodal framework for Protein-Protein Interaction prediction. By effectively integrating protein's inherent hierarchical organization: with sequence, structure, and functional information, MMM-PPI significantly improves prediction accuracy and generalization capabilities, particularly for unseen proteins and in data-scarce scenarios. The proposed method has the potential to accelerate drug discovery and synthetic biology workflows by providing a cost-effective, high-throughput alternative to experimental screening. Besides, the model's ability to identify high-attention motifs offers interpretability, aiding researchers in understanding the molecular mechanisms driving interactions and identifying novel therapeutic targets for disease treatment.

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

## A. Motif Detection and Statistics

**Motif Identification.** Motif occurrences are identified using the FIMO (Find Individual Motif Occurrences) algorithm (Grant et al., 2011) from the MEME Suite (Bailey et al., 2009). We use the PROSITE database (Version 2021_04) as the source of position-specific scoring matrices (PSSMs). Since our objective is not on de novo motif discovery, verifying the existence of motifs outside this established biological knowledge base falls beyond the scope of this work.

To ensure statistical significance, motif scanning is performed with a $p$-value threshold of $1 \times 10^{-4}$. A zero-order background model derived directly from the character frequencies of the input protein sequences is employed to accurately estimate the null hypothesis distribution. Regarding overlap resolution, no non-maximum suppression (NMS) is applied. All identified motif instances satisfying the significance threshold are retained to preserve potentially overlapping functional sites.

We acknowledge that FIMO identifies motifs based on sequence patterns rather than direct 3D geometry. We adopt this strategy primarily for scalability, as explicit 3D geometric motif mining is often computationally prohibitive for large-scale datasets (e.g., millions of proteins in STRING). Crucially, while the identification of motifs is sequence-based, our method ensures their encoding is spatially informed. By aggregating residue embeddings derived from the structure-modality HGNN, which models spatial dependencies via radius and K-nearest neighbor edges, the resulting motif embeddings explicitly incorporate the 3D local environment and conformational context. This design effectively bridges the gap between linear sequence patterns and their spatial structural behaviors without sacrificing computational efficiency.

**Motif-Related Statistics.** Table 10 shows the distribution of the number of motifs per protein. As can be seen, the average is comparable across the three datasets (45-48) with similar standard deviations (~37). The majority of proteins (approximately 65–70%) fall within the 0–50 motifs range across all datasets.

*Table 10.* Statistics on the number of motifs per protein across three datasets. The table reports the mean, standard deviation, minimum, maximum number of motifs, and the distribution of proteins grouped by motif count ranges.

| Dataset | mean | std | min | max | #proteins w/ 0-50 motifs | #proteins w/ 50-100 motifs | #proteins w/ 100-150 motifs | #proteins w/ ≥150 motifs |
|---|---|---|---|---|---|---|---|---|
| SHS27k | 45.62 | 36.84 | 1 | 200 | 1120 | 407 | 98 | 38 |
| SHS148k | 47.41 | 37.49 | 1 | 200 | 3346 | 1309 | 291 | 136 |
| STRING | 45.34 | 36.74 | 1 | 200 | 10224 | 3544 | 821 | 363 |

Table 11 shows the distribution of the lengths of all extracted motifs. The average is consistently around 16 amino acids across all datasets, with similar standard deviation, minimum, and maximum. Most motifs fall within the 10–20 residue range, but short motifs (0–10 residues) and longer ones (20–30 or ≥30 residues) are also present in significant numbers.

*Table 11.* Statistics on the motif lengths across three datasets. The table includes the mean, standard deviation, minimum, maximum motif lengths, and the number of motifs falling into different length intervals.

| Dataset | mean | std | min | max | #motifs w/ length 0-10 | #motifs w/ length 10-20 | #motifs w/ length 20-30 | #motifs w/ length ≥ 30 |
|---|---|---|---|---|---|---|---|---|
| SHS27k | 16.29 | 7.78 | 4 | 54 | 16879 | 42381 | 11543 | 5060 |
| SHS148k | 16.41 | 7.86 | 4 | 54 | 53274 | 133451 | 36895 | 17292 |
| STRING | 16.43 | 7.95 | 4 | 54 | 152133 | 372021 | 102570 | 50735 |

Table 12 shows the the proportion of unseen proteins that share at least $k$ motifs with one seen protein on the SHS27k dataset. Specifically, let $\mathcal{P}_{\text{unseen}}$ be the set of unseen proteins in the test set, and $\mathcal{P}_{\text{seen}}$ be the set of seen proteins in the training set. For each pair $(P_{\text{unseen}}, P_{\text{seen}}) \in \mathcal{P}_{\text{test}} \times \mathcal{P}_{\text{train}}$, we obtain the number of shared motifs $|M(P_{\text{unseen}}) \cap M(P_{\text{seen}})|$, where $M(P)$ is the set of motifs in protein $P$. As can be seen from Table 12, (i) All unseen proteins share at least one shared motif ($k = 1$) with a seen protein. (ii) Although reducing the label ratio leads to sharing with fewer seen proteins, even with a label ratio as low as 5%, still more than half of the unseen proteins share a relatively high number of motifs ($k = 10$) with at least one seen protein. Overall, this demonstrates the significant degree of motif overlap between unseen and seen proteins, thus supporting the idea that motif sharing is a key factor in the model's generalization ability on unseen proteins. we evaluate how many motifs each unseen protein in the test set shares with each seen protein in the training set.

*Table 12.* Proportion (%) of unseen proteins (on the SHS27k dataset) sharing at least $k$ motifs with one seen protein.

| label ratio | random | | | | dfs | | | | bfs | | | |
|---|---|---|---|---|---|---|---|---|---|---|---|---|
| | $k=1$ | $k=5$ | $k=10$ | $k=20$ | $k=1$ | $k=5$ | $k=10$ | $k=20$ | $k=1$ | $k=5$ | k=10 | $k=20$ |
| **100%** | 100 | 89.68 | 63.49 | 16.67 | 100 | 91.71 | 56.68 | 15.21 | 100 | 89.10 | 53.38 | 14.66 |
| **20%** | 100 | 92.63 | 66.07 | 20.54 | 100 | 90.31 | 51.05 | 13.87 | 100 | 89.36 | 55.20 | 14.60 |
| **5%** | 100 | 86.91 | 50.70 | 13.65 | 100 | 89.68 | 50.28 | 13.32 | 100 | 87.50 | 52.14 | 13.93 |

## B. Details of PPI Prediction

Following (Lv et al., 2021; Zhao et al., 2023; Gao et al., 2023; Wu et al., 2024), we utilize the PPI graph $\mathcal{G} = (\mathcal{P}, \mathcal{X})$(resp. $\mathcal{G} = (\mathcal{P}, \mathcal{X}_\mathcal{K})$) for the transductive (resp. inductive) setting, and adopt the Graph Isomorphism Network (GIN) (Xu et al., 2019) and a linear layer to predict the PPI labels. Following (Wu et al., 2024), we use the pre-trained encoder as an off-the-shelf tool to encode the pair of proteins $(P_a, P_b)$ in the PPI set $\mathcal{X}$. Due to the pair-wise encoding process, each protein $P_a$ has $D_a$ embeddings, where $D_a$ is the number of protein pairs that $P_a$ appears. We average the $D_a$ embeddings to get the unique embedding $\hat{\mathbf{h}}_a$. To ensure evaluation integrity in the inductive setting, we enforce a strict separation. During training, the initial node embeddings are aggregated from interactions only present in the training set $\mathcal{X}_K$, ensuring that no context from unseen test pairs leaks into the GNN optimization. During inference, the MMM-PPI encoder utilizes all the pairs in $\mathcal{X}$ to generate context-aware node features.

The update of the node features at the $l$-th layer ($1 \leq l \leq L_{GIN}$) is:

$$\mathbf{w}_i^{(l)} = \text{FCL}^{(l)}\left(\left(1 + \epsilon^{(l)}\right) \cdot \mathbf{w}_i^{(l-1)} + \sum_{x_{i,j} \in \mathcal{X}} \mathbf{w}_j^{(l-1)}\right),$$

where $\epsilon^{(l)}$ is a learnable parameter, $\mathbf{w}_i^{(0)} = \hat{\mathbf{h}}_i$, and $FCL^{(l)}$ is a fully connected layer of the $l$-th layer.

To predict the interaction between $P_a$ and $P_b$, The dot product of $\mathbf{w}_a^{(L_{GIN})}$ and $\mathbf{w}_b^{(L_{GIN})}$ is fed into a FCL layer to predict their interaction type $\hat{\mathbf{y}}_{ij} = \text{FCL}(\mathbf{w}_a^{(L_{GIN})} \cdot \mathbf{w}_b^{(L_{GIN})})$. Finally, the binary cross-entropy is minimized to train the GIN and PPI predictor.

---

**Algorithm 1** Pre-training the Hierarchical Motif-based Multimodal Encoder.

---

**Require:** Protein set $\mathcal{P}$, Training PPI pairs $\mathcal{X}_K$, number of pre-training epochs $E_{pre}$.
 1: **# 1. Pre-processing**
 2: **for** modality $\in \{\text{sequence}, \text{structure}, \text{function}\}$ **do**
 3:     Encode residue embeddings for each protein;
 4: **end for**
 5: Identify motifs $\{M_{i_1}, \ldots, M_{i_k}\}$ for each protein $P_i$ via FIMO;
 6: **# 2. Hierarchical Encoder Optimization**
 7: Initialize the multimodal motif encoder and protein encoder parameters $\Theta_{Enc}$.
 8: **for** epoch $= 1, 2, \ldots, E_{pre}$ **do**
 9:     **for** each batch of protein pairs $(P_a, P_b) \in \mathcal{X}_K$ **do**
10:         *// Micro-scale*
11:         Concatenate embeddings from all three modalities into multimodal residue embeddings;
12:         *// Meso-scale*
13:         Aggregate residues to obtain motif embeddings $H_a, H_b$;
14:         *// Macro-scale*
15:         Generate context-aware embeddings $\tilde{\mathbf{h}}_{a|b}$ and $\tilde{\mathbf{h}}_{b|a}$ via pairwise motif co-attention;
16:         Predict interaction score $\hat{y}_{ab} = \text{FCL}(\text{concat}[\tilde{\mathbf{h}}_{a|b}, \tilde{\mathbf{h}}_{b|a}])$;
17:         Update $\Theta_{Enc}$ by minimizing the binary cross-entropy loss.
18:     **end for**
19: **end for**
20: **return** Pre-trained Encoder parameters $\Theta_{Enc}^*$.

---

## C. Pseudo-code of MMM-PPI

The pseudo-code of the proposed MMM-PPI is shown in Algorithm 1.

## D. Details of the Datasets

Following (Lv et al., 2021; Zhao et al., 2023; Gao et al., 2023; Wu et al., 2024), we use multi-type PPI annotations derived from the STRING database[2] (Szklarczyk et al., 2019). Each PPI is annotated with seven interaction types: Activation, Binding, Catalysis, Expression, Inhibition, Post-translational Modification (Ptmod), and Reaction. Following (Chen et al., 2019), we create two more challenging subsets, SHS27k and SHS148k, by randomly selecting proteins with more than 50 amino acids and less than 40% sequence identity from the Homo sapiens subset of STRING. Statistics of the three datasets are shown in Table 1.

To split the dataset into training, validation, and test sets, we adopt three splitting strategies from (Lv et al., 2021): Random, Breadth-First Search (BFS), and Depth-First Search (DFS). The Random splitting method follows the simple approach of randomly assigning data to the training, validation, and test sets. The BFS strategy simulates the scenario where unknown proteins are densely interconnected, forming clusters within the PPI network. In contrast, the DFS strategy is designed to simulate the scenario where unknown proteins are sparsely distributed across the network, with limited interactions between them. To apply the BFS and DFS strategies, we first set the desired test set size and randomly select a root node in the PPI network. Then, using the corresponding strategy (BFS or DFS), we traverse the network starting from the root node. Proteins visited during the search are added to the test set, simulating the scenario of unknown proteins. The search process stops when the test set reaches the predetermined size.

We adopt both the transductive and inductive settings for PPI prediction. In the transductive setting, the GNN is applied on the graph constructed using the full dataset (including the training, validation, and test sets) and trained using the training labels. This approach allows the model to exploit the full structure of the graph. However, it assumes that the test set is fixed and available during training, which limits its ability to generalize to unseen data. In contrast, the inductive setting simulates real-world scenarios where the model is expected to generalize to unseen nodes. The model is applied on a subset of the graph (only the training set) and is asked to predict labels on edges that are not part of the training graph. This approach is more suitable for applications where the model needs to handle new, unseen data (such as dynamic or evolving graphs).

## E. Additional Ablation Studies

**Without using Pre-Trained Models.** In this experiment, we remove the pre-trained models used to learn multimodal residue embeddings in Section 3.1.1. Instead, we initialize the protein graph using the same embedding in (Wu et al., 2024; Gao et al., 2023).

*Table 13.* Model performance (Micro-F1 score, %)) without using pre-trained models. The training setting is transductive.

| Method | SHS27k | | | SHS148k | | | STRING | | | Avg. |
|---|---|---|---|---|---|---|---|---|---|---|
| | Random | DFS | BFS | Random | DFS | BFS | Random | DFS | BFS | |
| DPPI | 70.45 | 43.69 | 43.87 | 76.10 | 51.43 | 50.80 | 92.49 | 63.41 | 54.41 | 60.74 |
| DNN-PPI | 75.18 | 48.90 | 51.59 | 85.44 | 56.70 | 54.56 | 81.91 | 61.34 | 51.53 | 63.02 |
| PIPR | 79.59 | 52.19 | 47.13 | 88.81 | 61.38 | 58.57 | 93.68 | 64.97 | 53.80 | 66.68 |
| GNN-PPI | 83.65 | 66.52 | 63.08 | 90.87 | 75.34 | 69.53 | 94.53 | 84.28 | 75.69 | 78.17 |
| SemiGNN-PPI | 85.57 | 69.25 | 67.94 | 91.40 | 77.62 | 71.06 | 94.80 | 84.85 | 77.10 | 79.95 |
| HIGH-PPI | 86.23 | 70.24 | 68.40 | 91.26 | 78.18 | 72.87 | - | - | - | - |
| MAPE-PPI (without pre-training) | 87.54 | 73.93 | 74.23 | 92.16 | 82.15 | 75.26 | 95.57 | 86.68 | 78.67 | 82.91 |
| MMM-PPI (without pre-trained model) | **88.70** | **74.58** | **75.40** | **92.89** | **84.07** | **76.63** | **96.86** | **89.15** | **81.41** | **84.41** |

Table 13 compares the testing micro-F1 scores with baselines that also do not require additional pre-training in the transductive setting. As can be seen, despite the removal of the pre-trained knowledge, MMM-PPI still consistently outperforms all baselines. However, not surprisingly, its overall performance is lower than in its full version (in Table 2), confirming the benefit of leveraging pre-training.

**Ablation Studies in inductive Setting.** Table 14 shows the corresponding results in the inductive setting. As can be seen, in

---

[2]https://string-db.org/

*Table 14.* Ablation study (Micro-F1 score, %) on the contributions of multi-modal information (SEQ: sequence, STR: structure, FUNC: function), and the multimodal motif/protein encoder. The inductive setting is used.

| Row Index | Encoder | | Modality | | | SHS27k | | | SHS148k | | | STRING | | | Avg. |
|---|---|---|---|---|---|---|---|---|---|---|---|---|---|---|---|
| | motif | protein | SEQ | STR | FUNC | Random | DFS | BFS | Random | DFS | BFS | Random | DFS | BFS | |
| 1 | ✗ | ✗ | ✓ | ✗ | ✗ | 83.05 | 69.81 | 73.16 | 86.66 | 78.04 | 65.08 | 92.02 | 83.94 | 77.13 | 78.77 |
| 2 | ✓ | ✓ | ✓ | ✗ | ✗ | 86.93 | 71.44 | 73.75 | 91.08 | 79.41 | 66.54 | 95.07 | 85.23 | 74.23 | 80.41 |
| 3 | ✗ | ✗ | ✗ | ✓ | ✗ | 82.60 | 70.14 | 72.03 | 84.35 | 73.61 | 62.01 | 90.40 | 82.55 | 71.25 | 76.55 |
| 4 | ✓ | ✓ | ✗ | ✓ | ✗ | 86.12 | 71.64 | 72.68 | 91.50 | 76.00 | 62.60 | 94.64 | 85.28 | 75.58 | 79.56 |
| 5 | ✗ | ✗ | ✗ | ✗ | ✓ | 83.63 | 69.73 | 72.04 | 87.73 | 77.92 | 64.54 | 91.46 | 84.70 | 76.44 | 78.69 |
| 6 | ✓ | ✓ | ✗ | ✗ | ✓ | 86.85 | 70.49 | 73.73 | 91.33 | 79.30 | 66.87 | 94.85 | 85.97 | 74.99 | 80.49 |
| 7 | ✗ | ✗ | ✓ | ✓ | ✓ | 83.56 | 70.11 | 73.41 | 86.07 | 78.00 | 65.26 | 91.10 | 84.07 | 73.44 | 78.34 |
| 8 | ✓ | ✗ | ✓ | ✓ | ✓ | 83.61 | 68.65 | 73.63 | 86.67 | 78.55 | 65.89 | 90.95 | 84.76 | 75.28 | 78.67 |
| 9 | ✗ | ✓ | ✓ | ✓ | ✓ | 86.10 | 72.02 | 74.46 | 87.76 | 81.11 | 66.90 | 93.51 | 86.82 | 75.87 | 80.51 |
| 10 | ✓ | ✓ | ✓ | ✓ | ✓ | **87.27** | **73.82** | **75.63** | **92.04** | **82.33** | **67.34** | **95.69** | **87.87** | **76.53** | **82.06** |

*Table 15.* Ablation study (Micro-F1 score, %) on the effectiveness of using three modality-specific HGNNs.

| Setting | model | SHS27k | | | SHS148k | | |
|---|---|---|---|---|---|---|---|
| | | Random | DFS | BFS | Random | DFS | BFS |
| trans-ductive | early fusion | 86.01 | 73.41 | 72.19 | 91.68 | 81.08 | 73.96 |
| | one HGNN | 88.60 | 74.60 | 75.87 | 91.54 | 82.64 | 74.87 |
| | three HGNN | **89.25** | **75.93** | **79.09** | **93.38** | **84.24** | **77.70** |
| in-ductive | early fusion | 85.80 | 70.85 | 71.53 | 89.27 | 79.75 | 64.28 |
| | one HGNN | 86.21 | 71.28 | 72.75 | 90.95 | 80.52 | 65.74 |
| | three HGNN | **87.27** | **73.82** | **75.63** | **92.04** | **82.33** | **67.34** |

*Table 16.* Ablation study (Micro-F1 score, %) on the comparison of different aggregation methods.

| Setting | fusion method | SHS27k | | | SHS148k | | |
|---|---|---|---|---|---|---|---|
| trans-ductive | sum | 88.71 | 74.21 | 78.28 | 92.98 | 83.25 | 77.01 |
| | MLP | 88.87 | 74.77 | 78.01 | 93.16 | 83.88 | 77.08 |
| | concat | **89.25** | **75.93** | **79.09** | **93.38** | **84.24** | **77.70** |
| in-inductive | sum | 85.53 | 70.42 | 74.15 | 90.48 | 79.54 | 64.97 |
| | MLP | 85.79 | 71.79 | 75.06 | 91.10 | 80.81 | 66.81 |
| | concat | **87.27** | **73.82** | **75.63** | **92.04** | **82.33** | **67.34** |

the inductive setting, both motif encoder, protein encoder, and multi-modal integration are still critical in enhancing protein embedding learning. These findings highlight the robustness of our method across different evaluation settings.

**Using One or Shared HGNN.** As the three modalities contain relatively orthogonal PPI information, using a separate heterogeneous graph neural network (HGNN) for each modality can enable the model to capture modality-specific information more effectively. To validate the benefit of this design, we conduct an ablation study by comparing it against two alternative configurations: (i) early fusion, where features from all three modalities are concatenated before being processed by a single HGNN; and (ii) shared HGNN, where all modalities are jointly processed using a shared HGNN.

Table 15 shows the Micro-F1 scores for the various configurations. As can be seen, the model using separate HGNNs consistently achieves the highest Micro-F1 scores, confirming that leveraging modality-specific HGNNs significantly enhances model performance.

**Using Different Aggregation Methods.** To evaluate the effectiveness of different information aggregation strategies across modalities in the multimodal motif encoder, we conduct an ablation study comparing three fusion methods: (i) element-wise summation (sum), (ii) a two-layer multilayer perceptron (MLP), and (iii) feature concatenation (concat).

Table 16 shows the testing micro-F1 scores for the various configurations. As can be seen, concatenation-based aggregation used in the proposed MMM-PPI model consistently achieves superior performance compared to summation and MLP-based fusion. Although MLP is more expressive, it may suffer from overfitting. In contrast, feature concatenation preserves the raw information of each modality's representation, enabling the pair-wise co-attention mechanism of the MMM-PPI model to learn cross-modal interactions more effectively.

**Ablation of Different Motif Lengths.** To further assess the impact of motif length on model performance, we conduct an ablation study in which only a subset of motifs within specific length intervals are used during training. Table 17 shows the Micro-F1 scores on the SHS27k and SHS148k datasets.

As can be seen, using motifs of all lengths ("all") consistently yields the highest performance, confirming that motifs of

*Table 17.* Ablation study(Micro-F1 score, %) on the contribution of different motif lengths.

| Setting | Motif length | SHS27k | | | SHS148k | | |
|---|---|---|---|---|---|---|---|
| | | Random | DFS | BFS | Random | DFS | BFS |
| trans-ductive | 0-10 | 87.95 | 73.52 | 76.30 | 92.23 | 82.98 | 75.05 |
| | 10-20 | 88.84 | 74.73 | 77.75 | 93.29 | 83.17 | 76.76 |
| | 20-30 | 88.02 | 73.58 | 76.22 | 93.20 | 82.81 | 76.64 |
| | $\geq$30 | 89.04 | 73.87 | 77.52 | 93.23 | 83.56 | 77.41 |
| | all | **89.25** | **75.93** | **79.09** | **93.38** | **84.24** | **77.70** |
| in-ductive | 0-10 | 84.45 | 67.55 | 71.20 | 90.77 | 80.31 | 64.22 |
| | 10-20 | 86.23 | 71.06 | 74.09 | 91.62 | 81.07 | 66.31 |
| | 20-30 | 85.32 | 66.91 | 72.13 | 91.58 | 79.66 | 65.29 |
| | $\geq$30 | 86.25 | 70.39 | 74.22 | 91.86 | 81.31 | 66.43 |
| | all | **87.27** | **73.82** | **75.63** | **92.04** | **82.33** | **67.34** |

*Table 18.* AUC scores(%) of various methods on different datasets and partitions with the transductive and inductive setting.

| Setting | method | SHS27k | | | SHS148k | | | STRING | | |
|---|---|---|---|---|---|---|---|---|---|---|
| | | Random | DFS | BFS | Random | DFS | BFS | Random | DFS | BFS |
| transductive | ESM2-3b | 95.36 | 84.18 | 86.61 | 97.94 | 92.63 | 86.18 | 99.01 | 94.83 | 89.70 |
| | MAPE-PPI | 94.81 | 83.81 | 85.98 | 97.66 | 92.00 | 87.12 | 98.94 | 95.57 | 89.83 |
| | KeAP | 95.63 | 85.21 | 87.39 | 97.28 | 92.24 | 87.34 | 98.08 | 95.29 | 88.42 |
| | MicroEnvPPI | 95.45 | 83.28 | 87.42 | 96.79 | 91.52 | 87.20 | 98.14 | 94.47 | 87.75 |
| | MASSA | 95.93 | 84.83 | 87.74 | 97.22 | 91.85 | 87.61 | 98.28 | 95.03 | 87.46 |
| | MMM-PPI | **96.10** | **86.74** | **89.14** | **98.03** | **93.57** | **88.50** | **99.34** | **96.85** | **91.21** |
| inductive | ESM2-3b | 89.43 | 80.31 | 86.67 | 96.11 | 91.61 | 78.62 | 97.72 | 91.78 | 86.64 |
| | MAPE-PPI | 88.59 | 80.51 | 84.24 | 94.60 | 89.78 | 70.11 | 96.91 | 92.60 | 84.10 |
| | KeAP | 92.96 | 81.78 | 85.76 | 96.48 | 91.41 | 78.69 | 97.53 | 92.30 | 85.11 |
| | MicroEnvPPI | 93.60 | 81.63 | 86.60 | 96.62 | 90.91 | 75.52 | 97.26 | 92.82 | 84.33 |
| | MASSA | 92.76 | 82.74 | 86.66 | 96.75 | 91.04 | 76.65 | 97.41 | 92.39 | 85.16 |
| | MMM-PPI | **95.89** | **85.57** | **88.17** | **97.68** | **92.87** | **80.91** | **98.74** | **94.57** | **87.81** |

varying lengths contribute complementary information beneficial to the prediction task. Among the subset length settings, motifs in the range of 10–20 and those with lengths $\geq$30 show notably strong performance. This suggests two possible factors: (i) motifs of length 10–20 are more prevalent in the dataset, providing a richer signal during training, and (ii) longer motifs ($\geq$30) may capture more complex structural or semantic information, enhancing the model's ability to generalize.

## F. More Evaluation Metrics

In addition to the Micro-F1 scores reported in Table 2, we further present the Area Under the ROC Curve (AUC) scores in Table 18 and the Area Under the Precision-Recall Curve (AUPR) scores in Table 19 to enable a more comprehensive performance comparison across models.

Consistent with the results in the Table 2, the proposed MMM-PPI model outperforms all baseline methods, achieving the highest AUC and AUPR scores across all datasets. These results further substantiate the robustness and effectiveness of MMM-PPI in encoding high quality protein embedding for protein–protein interaction tasks.

## G. Implementation Details

All experiments are based on PyTorch 1.12.0 and CUDA 11.6, with 256 AMD EPYC 7763 64-Core Processor and 4 NVIDIA A100 GPUs. For all the datasets and partitions, we use the following hyperparameter settings: number of pre-training epochs $E_{pre} = 50$, pre-training learning rate 0.0005, pre-training weight decay 0.0005, PPI training epoch $E_{ppi} = 500$, PPI encoder (GIN) with $L_{GIN} = 2$ layers and hidden dimension of 1024. The other dataset- or partition-specific hyperparameters are determined using the AutoML toolkit NNI[3] with the following search space: modality-specific HGNN embedding dimension: $\{128, 256, 512\}$; number of layers in modality-specific HGNN $\{1, 2\}$; pair-wise motif co-attention dimension $U = \{128, 256, 512\}$; and dropout ratio $\{0.1, 0.2\}$.

---

[3]https://github.com/Microsoft/nni

*Table 19.* AUPR scores(%) of various methods on different datasets and partitions with the transductive and inductive setting.

| Setting | method | SHS27k | | | SHS148k | | | STRING | | |
|---|---|---|---|---|---|---|---|---|---|---|
| | | Random | DFS | BFS | Random | DFS | BFS | Random | DFS | BFS |
| transductive | ESM2-3b | 88.86 | 68.57 | 75.25 | 92.76 | 83.19 | 76.91 | 91.23 | 84.18 | 73.78 |
| | MAPE-PPI | 89.13 | 66.97 | 72.63 | 92.32 | 82.46 | 77.79 | 91.21 | 85.73 | 74.42 |
| | KeAP | 88.84 | 68.16 | 76.64 | 92.50 | 83.19 | 77.91 | 90.54 | 84.67 | 75.08 |
| | MicroEnvPPI | 89.70 | 68.25 | 77.20 | 92.29 | 83.57 | 77.73 | 90.41 | 84.98 | 74.09 |
| | MASSA | 89.37 | 67.86 | 77.43 | 91.89 | 82.85 | 78.03 | 90.95 | 85.47 | 72.04 |
| | MMM-PPI | **90.07** | **69.71** | **78.34** | **93.14** | **84.75** | **79.26** | **91.68** | **86.39** | **76.45** |
| inductive | ESM2-3b | 79.40 | 62.64 | 74.04 | 89.60 | 80.45 | 67.69 | 86.90 | 77.77 | 66.97 |
| | MAPE-PPI | 78.16 | 62.95 | 70.36 | 86.90 | 79.03 | 64.92 | 85.03 | 78.36 | 64.04 |
| | KeAP | 86.62 | 66.09 | 74.46 | 90.42 | 81.82 | 62.91 | 86.78 | 78.75 | 65.50 |
| | MicroEnvPPI | 85.23 | 65.83 | 75.43 | 90.71 | 81.60 | 64.98 | 86.05 | 79.06 | 65.27 |
| | MASSA | 84.68 | 64.21 | 72.89 | 91.03 | 80.73 | 63.14 | 85.29 | 79.64 | 66.46 |
| | MMM-PPI | **88.91** | **71.56** | **76.74** | **92.36** | **83.43** | **66.52** | **88.06** | **80.22** | **68.08** |

*Table 20.* Biological function correspondence between the motifs with high attention scores assigned by MMM-PPI and the protein-protein interaction type.

| | |
|---|---|
| **Protein pairs** | (ENSP00000000233, ENSP00000006101) |
| **PPI Type** | reaction, catalysis |
| **High-attention motifs** | ATP GTP A (in ENSP00000000233), ADENYLATE KINASE (in ENSP00000006101) |
| **Biological correspondence Interpretation** | The motif ATP GTP A in ENSP00000000233 is associated with nucleotide binding, particularly ATP/GTP, which is critical for energy transfer and signal transduction. The motif ADENYLATE KINASE in ENSP00000006101 facilitates the reversible transfer of phosphate groups among adenine nucleotides (e.g., ATP, ADP, AMP). These molecular functions directly contribute to catalytic processes and nucleotide cycling, aligning well with the annotated interaction types involving reaction and catalysis. |
| **Protein pairs** | (ENSP00000000412, ENSP00000223023) |
| **PPI Type** | reaction, catalysis, binding |
| **High-attention motifs** | CASPASE CYS (in ENSP00000000412), HOK GEF (in ENSP00000223023) |
| **Biological correspondence Interpretation** | The motif CASPASE CYS in ENSP00000000412 refers to the cysteine residue within the active site of caspase enzymes. Caspases are a family of cysteine proteases that play a crucial role in apoptosis (programmed cell death) and inflammation. The motif HOK GEF in ENSP00000223023 refers to a bacterial toxin-antitoxin system component, known to disrupt membrane integrity and induce programmed cell death. The shared involvement of these motifs in apoptosis, cytotoxic processes, and programmed cell death supports the observed PPI characterized by reaction, catalysis, and binding, suggesting a functional alignment in cell death pathways. |
| **Protein pairs** | (ENSP00000229595, ENSP00000262133) |
| **PPI Type** | activation |
| **High-attention motifs** | ALDOLASE CLASS II 2 (in ENSP00000229595), CHAPERONINS CPN60 (in ENSP00000262133) |
| **Biological correspondence Interpretation** | The motif ALDOLASE CLASS II 2 in ENSP00000229595 is an enzyme that plays a crucial role in carbohydrate metabolism, specifically in the breakdown and synthesis of sugars. It's characterized by its use of divalent metal ions (like zinc) to catalyze the cleavage or condensation of carbon-carbon bonds in sugar molecules. The motif CHAPERONINS CPN60 in ENSP00000262133 is linked to molecular chaperones that mediate correct protein folding, particularly under stress conditions. The interaction between a metabolic enzyme and a chaperone protein suggests a regulatory activation relationship, where chaperonin activity could be essential for maintaining the functionality of metabolic enzymes, thus supporting the annotated "activation" PPI. |

# H. Qualitative Analysis

Although MMM-PPI demonstrates state-of-the-art performance in quantitative evaluations, it is crucial to assess the biological interpretability of the learned representations. Table 20 shows three example protein–protein interactions, with (i) the PPI interaction type, (ii) motifs that are assigned the highest attention scores by MMM-PPI for each protein pair, and (iii) biological interpretation describing how the motifs are functionally related to the interaction type.

As can be seen, across all three examples, there is a strong alignment between the functional roles of high-attention motifs and nature of the corresponding PPIs. This demonstrates that the motifs receiving high attention from MMM-PPI are not only statistically significant but also biologically meaningful, thereby validating the model's ability to learn interpretable, biologically grounded features.

## I. Statistical Significance of Performance Gains

To verify the statistical significance of the improvements reported in Table 2, we conduct a paired bootstrap test against the strongest baseline MicroEnvPPI. We resample the test sets with replacement for 1,000 iterations and evaluate both models on each resampled set to compute the paired difference $\Delta_{\text{Micro-F1}} = \text{MMM-PPI} - \text{MicroEnvPPI}$. Table 21 reports the 95% confidence intervals (CI) and $p$-values under the transductive setting. All $p$-values are below $0.05$ and the lower bounds of all 95% CIs are strictly positive, confirming that the improvements of MMM-PPI are statistically significant.

*Table 21.* Paired bootstrap test (1,000 iterations) comparing MMM-PPI with MicroEnvPPI under the transductive setting.

| Dataset | Split | Paired $\Delta$ | 95% CI | $p$-value |
|---------|-------|------------------|--------|-----------|
| SHS27k | Random | 0.87 | [0.05, 1.58] | 0.019 |
|        | DFS    | 1.43 | [0.26, 2.66] | 0.007 |
|        | BFS    | 2.80 | [2.12, 4.27] | <0.001 |
| SHS148k | Random | 0.47 | [0.28, 0.66] | <0.001 |
|         | DFS    | 1.15 | [0.80, 1.50] | <0.001 |
|         | BFS    | 1.26 | [1.01, 1.95] | <0.001 |

## J. Complementarity of Three Modalities

To further demonstrate the complementarity of the three modalities, we examine the overlap of correctly predicted test samples across the models using SEQ, STR, and FUNC individually (rows 2, 4, 6 in Table 4). Table 22 shows the percentage distribution of test samples with respect to the single-modality models that predict them correctly. As can be seen, while the majority of test samples can be correctly predicted by all three single-modality models, there is a notable portion of more difficult samples that can only be correctly predicted by one of the three single-modality models. This highlights the relatively orthogonal contributions of the sequence, structure, and function modalities to PPI prediction.

*Table 22.* Percentages of labels that are correctly predicted by the three single-modality models.

| Modality | SHS27k | | | | SHS148k | | | |
| | transductive | | inductive | | transductive | | inductive | |
| | DFS | BFS | DFS | BFS | DFS | BFS | DFS | BFS |
|----------|------|------|------|------|------|------|------|------|
| none | 9.83 | 6.69 | 5.61 | 5.62 | 6.78 | 9.52 | 6.36 | 10.67 |
| only SEQ | 1.59 | 2.82 | 3.85 | 2.30 | 1.31 | 1.90 | 2.12 | 7.33 |
| only STR | 2.33 | 2.36 | 4.29 | 4.16 | 1.55 | 2.49 | 2.24 | 4.13 |
| only FUNC | 2.73 | 2.64 | 3.91 | 3.34 | 1.14 | 1.62 | 2.12 | 5.10 |
| SEQ and STR | 4.52 | 3.54 | 6.59 | 6.50 | 1.91 | 3.19 | 3.32 | 5.98 |
| SEQ and FUNC | 4.34 | 3.94 | 7.65 | 6.75 | 1.85 | 3.10 | 3.41 | 9.39 |
| STR and FUNC | 3.67 | 2.84 | 6.82 | 5.79 | 1.55 | 2.03 | 2.66 | 8.24 |
| all 3 modalities | 70.97 | 75.16 | 61.27 | 65.54 | 83.91 | 76.16 | 77.77 | 49.16 |

