# OpenReview forum: "Enhancing Protein-Protein Interaction Prediction with Hierarchical Motif-based Multimodal Protein Embedding"
_ICML.cc/2026/Conference — ICML 2026 regular_

### Official Review · Reviewer_v2CY · 2026-03-04

**Soundness:** 3
**Presentation:** 4
**Significance:** 2
**Originality:** 2
**Overall Recommendation:** 3
**Confidence:** 4

**Summary:**

This paper introduces MMM-PPI, an enhanced protein representation learning framework that boosts downstream protein-protein interaction (PPI) prediction. The key innovation lies in a hierarchical design that bridges micro-level residue embeddings and macro-level protein representations through meso-level motif information. A novel pairwise context-aware motif co-attention mechanism assigns protein-specific weights to motifs, enabling more accurate final protein representations. Additionally, starting from the construction of micro-level residue embeddings, the framework integrates complementary information across three modalities: sequence, function, and structure. Comprehensive experiments demonstrate that MMM-PPI-based models outperforms previous models in PPI prediction under challenging data partitions and limited training data.

**Compliance With Llm Reviewing Policy:**

Affirmed.

**Final Justification:**

After carefully inspecting the author rebuttal reply, I am still not convinced by the originality of the proposed PPI modeling framework. In specific, the atomic-level interactions between proteins, a key determinant of PPI, is not well captured by the proposed model, which limits its potential towards physically plausible predictions. Therefore, I choose to keep my rating unchanged.

**Key Questions For Authors:**

1. Given the large scale of the dataset, the case studies in appendix analyze only three high-attention examples. Is there a broader, statistically significant correspondence between high-attention motifs and known biological functions across the entire dataset? Furthermore, do motifs that receive low attention scores consistently lack biological relevance, or could this indicate a potential limitation in the attention mechanism's ability to capture all functionally significant sites?
2. Since MMM-PPI relies on FIMO doing sequence motif identification, how’s MMM-PPI’s robustness when motif detection is affected by noise or omission?

**Limitations:**

No, the paper does not include a dedicated section discussing the limitations of the methods and their potential negative impacts. A more detailed discussion on the model's limitations would be beneficial, such as the trade-off in computational resources when scaling to full modalities.

**Strengths And Weaknesses:**

Strengths
1. Clear and well-justified scientific motivation: The authors clearly identify two key limitations in existing PPI prediction methods: the insufficient modeling of proteins’ hierarchical organization and the reliance on coarse-grained multimodal fusion strategies. This problem formulation is well grounded in biological principles, particularly motif-driven binding mechanisms, thereby providing strong biological plausibility for the proposed approach. Furthermore, several case studies demonstrate that the identified high-attention motifs exhibit meaningful biological correspondence, supporting the interpretability of the model’s predictions.
2. The hierarchical modeling framework is well-designed and comprehensively considers multimodal features: MMM-PPI forms a clear bottom-up pipeline, comprising micro-scale residue multimodal encoding, meso-scale motif construction, and macro-scale protein co-attention aggregation. The deliberate use of motifs as meso-scale representational bridges   avoids direct residue pooling and aligns the model's design with the actual granularity of PPI binding. This represents a conceptual improvement over numerous GNN-based PPI works.
3. Scalability of the framework: MMM-PPI effectively leverages multiple pretrained models, such as ESM, ensuring strong compatibility with existing foundation protein models while enabling flexible integration with future advances in protein representation learning.
4. Clarity of presentation: The paper presents a coherent narrative from problem formulation to solution design. Extensive experiments are conducted, and the results are clearly and systematically summarized through well-structured tables.

Weaknesses
1. Limited methodological novelty: The proposed approach appears to focus primarily on engineering integration, largely combining existing techniques and tools rather than introducing fundamentally new methodological innovations.
2. Unclear efficiency–performance trade-off: Although performance improvements are reported, they are relatively modest while the training cost increases substantially (by roughly 1.5x), raising concerns about the fairness of the trade-off. Moreover, only training efficiency is evaluated, whereas inference efficiency is not discussed. Given the complexity of the overall pipeline, it remains unclear whether the observed gains stem from the proposed hierarchical design or simply from increased model capacity, as parameter-controlled comparisons are lacking.

---

> ### Author Rebuttal · Authors · 2026-03-31
>
> We sincerely thank the reviewer for the comments.
>
> ### Weakness 1
>
> Please see the first reply to DD5U about novelty.
>
>
>
> ### Weakness 2
> **Significance of Performance Gains:** Note that the baselines are currently SOTA models. MMM-PPI achieves new SOTA results on all three datasets in both transductive and inductive settings. More importantly, it shows a clear advantage in the harder BFS and DFS splits, with a 1~ 2% improvement.
>
> **Parameter-Controlled Comparison:** We implemented a **parameter-matched flat model** by directly averaging the residue embedding in eq (3) to calculate the protein embeddings $P_a$ and $P_b$. A transformer backbone with a similar parameter size to MMM-PPI is then used to aggregate the features. The following are test results in the transductive setting.
>
>
> ||#parms|| SHS27k ||
> |-|-|-|-|-|
> |||Random|DFS|BFS|
> |parameter-matched model|23.69M|78.59|66.08|74.53|
> |MMM-PPI|23.68M|89.25|75.93|79.09|
>
> ||Encoder Pre-training||GNN Training||
> |-|-|-|-|-|
> ||time|memory|time|memory|
> |parameter-matched model|60:37|27000|00:55|1800|
> |MMM-PPI|59:58|38000|00:56|1800|
>
> Compared to MMM-PPI, this flat parameter-matched model has similar training time but a significantly lower Micro-F1 score. The flat model uses less memory because it performs simple averaging of the residue embeddings, while MMM-PPI needs more memory to encode the middle-level motifs for better accuracy.
>
> **Inference Efficiency:**
> Similar to training, the inference process also has two stages: protein encoder inference and GNN inference.  We show the inference time and memory consumption on the SHS27k test set in
> the following:
>
> ||Encoder Inference||GNN Inference||
> |-|-|-|-|-|
> ||time|memory|time|memory|
> |MAPE-PPI|35s|4400MiB|0.24s|380MiB|
> |MMM-PPI (structure)|47s|4100MiB|0.21s|370MiB|
> |capacity-matched model|1m28s|5000MiB|0.31s|480MiB|
> |MMM-PPI (full)|1m30s|14000MiB|0.26s|480MiB|
>
> Comparing MAPE-PPI to our MMM-PPI (structure), the hierarchy arch adds only 12 seconds to encoder inference and slightly reduces memory usage. The increased time and memory for the full MMM-PPI is the expected cost of executing the two additional modalities (sequence and function). For GNN inference, while the multi-modality models require slightly more memory than the single-modality models due to the larger multi-modal features, the overall cost remains lightweight.
>
>
>
> ### Question 1
>
> Notice that the case studies in appendix requires manually inspection, so it cannot be automated on the full dataset. To report a dataset-wide statistics as requested by the reviewer, we use a quantitative method following  "Paying attention to attention: High attention sites as indicators of protein
> family and function in language models. PLOS Computational Biology, 2025, 21(9)": check whether the animo sequence of the highest/lowest attention motifs overlaps with the UniProt-annotated biological function animo sequence. If they overlap by at least 1 amino acid, we call it a "Hit". For reference, we also add a random baseline that randomly assigns attention weights to the motifs within a protein. This exp is on SHS27k random splitting. The hit rates are as follows:
>
> ||Top-2 highest attention Motifs|Bottom-2 lowest attention Motifs|
> |-|-|-|
> |Random attention|24.36%|24.74%|
> |MMM-PPI|47.87%|15.60%|
>
> Our dataset-wide statistics show that high-attention motifs match biological functions in 47.87%, much more often than random choices in 24.36%, while low-attention motifs rarely match them, with 15.60% hit rate lower than random choices in 24.74%.
>
>
>
>
> ### Question 2
>
> FIMO detects motifs using PROSITE (Version 2021_04). Since the ground truth of motifs is unknown, FIMO's noise or omission cannot be measured. To test MMM-PPI's robustness, we simulated errors by randomly deleting motifs (simulating omission) and adding motifs (simulating noise). The transductive results are shown below.
>
> |||SHS27k|||SHS148k||
> |-|-|-|-|-|-|-|
> ||Random|DFS|BFS|Random|DFS|BFS|
> |MMM-PPI (omission 10%)|89.03|74.50|78.76|93.35|83.80|76.80|
> |MMM-PPI (omission 20%)|88.48|73.17|75.98|93.32|83.02|76.52|
> |MMM-PPI (omission 30%)|88.33|73.02|75.14|93.20|82.95|76.21|
> |MMM-PPI (noise 10%)|89.23|75.23|78.84|93.28|83.85|77.56|
> |MMM-PPI (noise 20%)|89.04|73.80|78.31|93.23|83.65|77.43|
> |MMM-PPI (noise 30%)|88.34|73.60|78.24|93.04|83.24|76.74|
> |MMM-PPI|89.25|75.93|79.09|93.38|84.24|77.70|
>
> The results show that MMM-PPI shows robust performance even under severe motif noise or omission.
>
>
>
> ### Limitations
>
> We will add the following limitation to the revised manuscript.
>
> The STRING and SHS datasets combine all types of proteins from whole organisms into a global network, rather than separating them by specific protein classes or tissue types. Therefore, a limitation of our model is that it predicts general protein interactions but cannot yet specify the exact cell type or tissue environment where these interactions occur.

---

> > ### Author Rebuttal · Reviewer_v2CY · 2026-04-01
> >
> > I appreciate the authors’ rebuttal, which addressed several of my concerns. In particular, the newly added parameter-controlled comparisons and inference efficiency analysis clarify that the performance gains are not solely due to increased model capacity.
> >
> > However, several concerns still remain partially resolved:
> >
> > 1.	Limited methodological novelty: The authors clarified their architectural innovations (hierarchical encoding and pairwise co-attention) and provided additional comparisons with ProtCLIP. However, the overall approach still largely integrates existing tools (ESM, FIMO, OntoProtein) and established techniques rather than introducing fundamentally new methodological principles. The rebuttal does not alter my assessment that the novelty is incremental.
> > 2.	Overly permissive definition of ‘Hit’: During analysis of high-attention motif interpretability, the authors consider any overlap of at least one amino acid between a motif and a UniProt-annotated functional region as a hit. Given that the majority of motif lengths in the dataset exceed 10, this one residue threshold is extremely low and prone to high false positive rates.
> > 3.	Limited robustness to motif detection errors: While the authors demonstrate overall stability, the performance drop under more severe perturbations is still non-negligible. For instance, under the motif omission 30% setting on the SHS27k dataset with the BFS split, the performance decreases by around 4 percentage points, suggesting that the robustness is not as strong as implied. Furthermore, randomly deleting or adding motifs is too idealized to capture the systematic biases inherent in real world motif detection.

---

> > > ### Author Response · Authors · 2026-04-02
> > >
> > > We appreciate the reviewer's continued engagement with our work.
> > >
> > > ### Q1
> > >
> > > We respectfully but firmly disagree that using existing tools (ESM, FIMO, OntoProtein) makes our novelty incremental. In the era of large foundation models, utilizing established models for basic feature extraction is the standard scientific practice.
> > >
> > > The core misunderstanding is equating the tools used for feature extraction with the methodology used for computation. Simply integrating these tools using traditional methods based on flat and static representations yields poor results, as shown by our baselines and ablation study. Our novelty lies in **fundamentally redesigning the computational architecture to match the actual biological mechanisms of protein interactions**.  In plain words, it is about how to use foundation models and tools.
> > >
> > > 1. **Structural Hierarchy (Moving away from "Flat" representations)**:  Previous methods operate on a "flat" principle, simply averaging all residue features into one static vector. Our methodology introduces a new computational principle: forcing the neural network to explicitly construct motif representations at the meso-scale level. This fundamentally changes the paradigm from a global average to a biologically meaningful composition of local functional units.
> > >
> > > 2. **Dynamic Interaction (Moving away from "Static" representations)**: Previous models assume a protein's representation is static. We introduce the methodological principle that a protein's representation must mathematically change based on the specific partner it is interacting with. Our pairwise co-attention is the mathematical implementation of this principle, dynamically calculating partner-specific interaction weights.
> > >
> > >
> > >
> > > ### Q2
> > >
> > > We thank the reviewer for this insightful point. While a 1-amino-acid overlap might risk false positives for longer motifs, enforcing a strict 10-amino-acid threshold would introduce severe false negatives from a biological perspective. Our dataset analysis shows that 80% of UniProt functional annotations (such as active sites, binding sites) are shorter than 5 amino acids.
> > >
> > > To balance rigor with biological reality, we replaced the absolute length threshold with an Annotation Coverage Ratio. A motif is now considered a "Hit" only if it covers at least 50% of the UniProt-annotated functional region.
> > >
> > > Under this stricter metric, our results are as follows:
> > > ||Top-2 highest attention Motifs|Bottom-2 lowest attention Motifs|
> > > |-|-|-|
> > > |Random attention|21.58%|21.60%|
> > > |MMM-PPI|41.89%|13.40%|
> > >
> > > As shown, the high-attention motifs consistently and significantly outperform the random baseline, demonstrating that MMM-PPI's attention mechanism aligns robustly with biological ground truth.
> > >
> > >
> > >
> > > ### Q3
> > >
> > > We sincerely thank the reviewer for the rigorous scrutiny.
> > >
> > > **A 4% drop under 30% data loss shows stability, not fragility.**
> > >
> > > A 30% motif omission is an extreme stress test. It destroys nearly one-third of the protein's motif information. The BFS split is a highly difficult setting because it tests entirely unseen proteins. In this setting, the model relies heavily on motifs to make predictions. When we randomly delete 30% of this critical input data, this means a severe collapse of the FIMO detection process, but the performance only drops by 4%. This proves that the model is highly stable. It does not easily collapse even under severe data corruption.
> > >
> > >
> > >
> > > **2. Addressing systematic bias with a different Motif Detection Algorithm.**
> > >
> > > We entirely agree that random deletion/addition is an idealized noise simulation and may not fully capture the systematic biases inherent in real-world motif detection tools.
> > >
> > > However, directly quantifying "systematic bias" is intrinsically challenging in this domain, as there is no ground truth for protein motifs. Therefore, to rigorously address the reviewer's concern regarding our model's robustness to systematic bias, we conducted an orthogonal validation by swapping the entire motif detection pipeline. Instead of FIMO + PROSITE (which relies on regular expression pattern matching), we evaluated MMM-PPI using HMMER + the Pfam database (which relies on profile Hidden Markov Models). This substitution completely shifts the underlying algorithmic assumptions and introduces a fundamentally different set of systematic biases.
> > >
> > > We re-trained and evaluated MMM-PPI. The transductive results are shown below:
> > >
> > > |Model|SHS27k(Random)|SHS27k(DFS)|SHS27k(BFS)|
> > > |-|-|-|-|
> > > |MicroEnvPPI (Best baseline)|88.28|74.57|76.22|
> > > |MMM-PPI (using FIMO + PROSITE)|89.25|75.93|79.09|
> > > |MMM-PPI (using HMMER + Pfam)|89.19|75.76|78.82|
> > >
> > > As the table shows, even when switching to a different motif detection algorithm with different systematic biases, MMM-PPI still significantly outperforms the best baseline, MicroEnvPPI. This proves that our performance gain comes from the hierarchical architecture itself and is highly robust to the choice and systematic biases of the upstream motif detection tools.

---

### Official Review · Reviewer_DD5U · 2026-03-11

**Soundness:** 4
**Presentation:** 2
**Significance:** 2
**Originality:** 2
**Overall Recommendation:** 3
**Confidence:** 5

**Summary:**

This paper proposes MMM-PPI, which is a Hierarchical Motif-based MultiModal protein Encoder for PPI Prediction. The authors incorporate representations across three dimensions, including micro-scale residues, meso-scale functional motifs, and macro-scale proteins. Based on comprehensive protein representations, the authors further develop the multimodal motif encoder and the multimodal protein encoder to facilitate protein-protein interaction prediction tasks. Extensive experiments on multiple benchmarks validate the superior performance.

**Compliance With Llm Reviewing Policy:**

Affirmed.

**Key Questions For Authors:**

1. The authors present a rigorous multidimensional integrated protein encoder tailored for PPI tasks, resulting in broad and substantial performance improvements. However, the rigor of the paper’s data processing details and the novelty of its research motivation remain relatively limited.
2. I would like to further ascertain whether the parameters of the protein encoder across the three modalities (sequence, function, and structure) are updated during training.
3. The authors aim to extract representations specifically targeting protein functions. OntoProtein is a foundational model that performs multimodal pretraining on protein sequences and functional textual descriptions based on knowledge graphs. This implies that OntoProtein has been exposed to protein–protein interaction (PPI) annotations during pretraining. If OntoProtein-derived protein representations are directly used for downstream PPI tasks, how do the authors ensure the absence of data leakage?
4. In the PPI prediction research area, there are studies that incorporate 3D structural information via spatial modeling to improve performance (e.g., SaProt [1]), as well as studies that leverage multimodal alignment to introduce functional information (e.g., ProtCLIP [2]). MMM-PPI merely combines three foundational information sources already known to benefit PPI prediction to achieve performance gains. Could the authors clearly articulate the innovative aspect of the study’s motivation?

[1] SaProt: Protein Language Modeling with Structure-aware Vocabulary. ICLR 2024 spotlight.

[2] ProtCLIP: Function-Informed Protein Multi-Modal Learning. AAAI 2025 oral.

**Limitations:**

Please provide a more holistic discussion of the work’s limitations and its potential negative societal impacts.

**Strengths And Weaknesses:**

Strengths:
1. The paper is clearly written, enabling readers to readily understand the model construction process.
2. The study’s findings are highly robust; MMM-PPI consistently achieves superior performance across a wide range of benchmarks.

Weaknesses:
1. Prior work has already integrated sequence, structural, and functional fragment information to optimize PPI performance, which limits the novelty of the present research study.
2. The overall framework of MMM-PPI is somewhat schematic, which constrains the model’s generalization capability in limited-information settings (e.g., when protein 3D structures are unavailable).

---

> ### Author Rebuttal · Authors · 2026-03-31
>
> We sincerely thank the reviewer for the comments.
>
> ### Novety(Weakness 1, Question 1, and Question 4)
>
> While using multiple data modalities is common, the core novelty of MMM-PPI is our specific model architecture.
> Existing models (including SaProt and ProtCLIP) learn flat and isolated representations by either global pooling or aligning individual proteins to texts,  **ignoring both the meso-scale motif information and the partner-dependent nature of interactions**.
>
> Our model has two main innovations to solve this:
>
> **Hierarchical Protein Encoding:** Instead of flat global pooling, we build a bottom-up hierarchy (Residue $\rightarrow$ **Motif** $\rightarrow$ Protein). This explicitly encodes motifs, which are fine-grained dependencies that regulate the protein’s behavior.
>
> **Pairwise Context-aware Co-attention:** We build a pairwise context-aware motif co-attention mechanism to capture the different motifs' action that depends on the specific partner protein  (Section 3.1.3, lines 210-213),  achieving dynamic calculation of the importance of Protein A's motifs based on Protein B.
>
> As suggested, we also added comparison with ProtCLIP using transductive setting and micro-F1 metric.
>
> |model||SHS27k|||SHS148k||
> |-|-|-|-|-|-|-|
> ||Random|DFS|BFS|Random|DFS|BFS|
> |ProtCLIP|88.44|73.31|76.07|92.83|83.04|76.39|
> |MMM-PPI|89.25|75.93|79.09|93.38|84.24|77.70|
>
> Table 2 and this result show that MMM-PPI performs much better than other multi-modal models (like SaProt, ProtCLIP). This proves that our performance gain comes from this specific hierarchical and context-aware design, not just from combining three data sources.
>
>
>
> ### Weakness 2
>
> We want to clarify how our model shows strong generalization capability with limited information:
>
> - **Missing 3D structures:** Our model uses predicted 3D coordinates from the AlphaFold database. Because AlphaFold2 predicts structures very well from just protein sequences, the problem of having zero 3D structure data is very rare today.
> - **Other limited information settings:** Our model still works very well in other limited settings. Section 4.5 shows that MMM-PPI beats all baselines even with only 5% of the training labels. Also, Table 4 shows that even if we use only one modality, our motif method still performs much better than standard models.
>
>
>
> ### Rigor of data processing (Question 1)
>
> Our data processing, data splitting, and evaluation follow the standard process in this research field. Details are in Section 4.1 and Appendix D:
>
> - **Dataset Construction:** We build the SHS27k and SHS148k datasets using a standard protocol from
>  "Multifaceted protein–protein interaction prediction based on siamese residual rcnn. Bioinformatics, 35(14), 2019".  We filter the STRING database to keep only proteins with > 50 amino acids and < 40% sequence identity. Details are in Appendix D.
> - **Graph Splitting:** We do not just use random splits. Following  "Learning unknown from correlations: Graph neural network for inter-novel-protein interaction prediction, arXiv:2105.06709,2021", we use Breadth-First Search (BFS) and Depth-First Search (DFS) to split the data into a 6:2:2 ratio. Details are in Appendix D.
>
>
>
> ### Question 2
>
> No, as stated in Section 3.1.1, line 156-157, the parameters of the pre-trained modality-specific protein models are not updated during our training process.
>
>
>
> ### Question 3
>
> We want to clarify that OntoProtein does not see PPI data during pre-training.
>
> - **No Data Leakage:** OntoProtein uses Gene Ontology (GO) to learn a single protein's function. The STRING database uses PPIs to show how proteins connect to each other. These are two different types of information, and so there is no data leakage.
> - **Standard Method:** Using GO to extract protein features in OntoProtein is a normal and accepted method in this field, just like the KeAP and MASSA.
> - **Proof from Results:** If OntoProtein already knew the PPI answers, its test scores would be very high. However, Table 2 shows that a model using only OntoProtein has a much lower Micro-F1 score than MMM-PPI. This proves that our good performance comes from our new model design (hierarchical protein encoding and pairwise context-aware co-attention), not from OntoProtein's pre-trained knowledge.
>
>
>
> ### Limitations
>
> Please see the reply to v2CY about limitations.
>
> #### Potential negative societal impacts
>
> There are some potential negative societal impacts. Highly accurate PPI models carry biosecurity risks, as they could be misused to design proteins that harm human health. Also, relying only on computer predictions without real lab tests can waste research time and money. MMM-PPI is designed to be a screening tool to guide real lab experiments, not to replace them.

---

> > ### Author Rebuttal · Reviewer_DD5U · 2026-04-04
> >
> > Thank the authors for their detailed response. Although most of my concerns have been addressed, the authors’ reply regarding the issue of pretraining data leakage has raised further concerns. Specifically, while OntoProtein may not have directly used PPI data during pretraining, the knowledge graph may contain annotated relations between protein nodes, and the attribute labels of these edges could include descriptions related to protein–protein interactions. If this issue has not been adequately recognized by the authors, it may significantly introduce a potential risk of data leakage.

---

> > > ### Author Response · Authors · 2026-04-04
> > >
> > > Thank you for your rigorous follow-up. We fully understand your concern. However, **as strictly defined in the original OntoProtein paper (Section 2.2 and Appendix A.1), we can confirm there is zero risk of PPI data leakage** because of the strict structure definition of the ProteinKG25 knowledge graph used by OntoProtein.
> > >
> > > 1. **There are NO Protein-Protein edges.** The pre-training graph contains exactly two types of edges:
> > >
> > > * **GO-to-GO:** Connecting two functional concepts to form a hierarchy.
> > > * **Protein-to-GO:** Connecting a single protein to a functional concept.
> > >
> > > There are absolutely **no edges connecting one protein directly to another protein**. The model never sees pairwise protein combinations during pre-training.
> > >
> > > 2. **Edge attributes describe functions, not physical interactions.**
> > >    The relations on the Protein-to-GO edges are purely functional (e.g., enables, part_of, involved_in). They only describe a single protein's biological role, not what it physically binds to. The edges on the GO-to-GO edges define the relationship between biological concepts. They do not state which specific protein interacts with which other specific protein.
> > >
> > > 3. As for multi-hop situations (e.g., Protein-to-GO-to-Protein), this path merely indicates that two proteins share a common biological attribute, which means functional similarity. This is fundamentally different from a direct physical connection. **Functional similarity does not equate to physical interaction.**
> > > 4. Furthermore, as stated in our rebuttal reply, empirical results confirm: **if OntoProtein actually suffered from ground-truth PPI data leakage, its baseline performance on downstream PPI tasks would be very high rather than so limited, and we cannot outperform OntoProtein significantly.**
> > >
> > > In short, because **OntoProtein is pre-trained on a graph completely lacking Protein-Protein edges and Protein-Protein interaction labels**, it is structurally impossible for it to leak ground-truth PPI data. We hope this clear breakdown of the knowledge graph fully resolves your concerns!

---

### Official Review · Reviewer_ya17 · 2026-03-13

**Soundness:** 3
**Presentation:** 3
**Significance:** 4
**Originality:** 4
**Overall Recommendation:** 5
**Confidence:** 3

**Summary:**

MMM-PPI proposes a hierarchical, multimodal protein encoder designed to improve the accuracy and interpretability of protein-protein interaction (PPI) prediction. The framework addresses two primary gaps in existing literature: the tendency of models to treat proteins as "flat" collections of residues—ignoring meso-scale functional motifs—and the under-utilization of complementary data from multiple biological modalities. The authors introduce a three-tiered, bottom-up encoding architecture:
- Micro-scale: Encoding residue features from sequence, structure, and function.
- Meso-scale: Aggregating these residues into spatially-informed motif embeddings using the FIMO algorithm.
- Macro-scale: Integrating motifs into global protein embeddings via a pairwise context-aware co-attention mechanism.

Extensive experiments on the STRING, SHS27k, and SHS148k datasets demonstrate that MMM-PPI outperforms state-of-the-art models, showing strong generalization to unseen proteins and robust performance in data-scarce scenarios.

**Compliance With Llm Reviewing Policy:**

Affirmed.

**Final Justification:**

I will retain overall recommendation, but I understand the concerns posed by the other reviewers. Mainly, the leakage issue posed by Reviewer U9xi and DD5U and the question about "hits" by Reviewer v2CY are aspects that need to thoroughly investigated for a camera-version (if the paper were to be accepted). Therefore, I have downgraded my confidence of my score from a 4 to a 3.

**Key Questions For Authors:**

1. Explainability: Can the method provide insight into why two specific proteins interact? Specifically, can you demonstrate if the model can learn interaction drivers through visualizations, integrated gradient importance scores, or other XAI methods?
SNP Consequence Assessment: Could this framework be utilized to assess if a Single Nucleotide Polymorphism (SNP) that alters an amino acid has measurable consequences for protein binding?
2. Shapley Value Analysis: To clarify what is driving the impressive performance, could you provide a Shapley value analysis on the ablation studies? This would involve evaluating the combinations of Encoder and Modality (e.g., $2^5-1=31$ evaluations) to assess the relative importance of each modality.
3. Generalization to De Novo Motifs: Given that Table 9 shows most "unseen" proteins share motifs with the training set, how does the model perform on truly "unseen motifs" or de novo motifs? This is critical to prove the model has learned generalized interaction rules rather than memorizing a lookup table of PROSITE IDs.
4. Novel Discovery: Are there specific interactions that MMM-PPI predicts with high certainty/attention scores that are currently missing from established databases but appear biologically promising upon interpretation?

**Limitations:**

The authors have discussed the sequence-based nature of FIMO and the computational costs of multimodal integration. However, the impact would be improved by providing a brief summary of the "types" of proteins in the STRING and SHS databases (e.g., categories like cytokines or transcription factors, and the specific tissue/cell types where interactions were studied). Authors should provide more clarity on the specific biological "kinds" of proteins and interactions their model is best suited to analyze.

**Strengths And Weaknesses:**

Soundness: The technical approach is grounded in established pre-trained models such as ESM2-3b, MAPE-PPI, and OntoProtein. The decoupling of the encoder pre-training from GNN training is a sensible design choice for computational efficiency. However, a potential technical concern arises regarding the "unseen" protein evaluation; since most unseen proteins in the test set contain motifs that appeared during training, the model may be performing a motif "lookup" rather than learning generalized biological rules.

Presentation: The paper is well-structured and the hierarchical levels of the model are clearly illustrated. The qualitative analysis in Appendix H provides useful biological grounding. The clarity could be further enhanced by providing high-level summaries of the "types" of proteins and interaction categories present in the primary databases used for training.

Significance: The work addresses a highly relevant problem in computational biology. Moving beyond flat representations to a motif-based hierarchy is a significant step for drug discovery and disease research. For instance, this framework could have a major impact on understanding how genetic mutations influence protein binding or cleavage—such as the cleavage of APP in Alzheimer's disease—where altered fragments have distinct biological properties.

Originality: The novelty lies in the hierarchical combination of sequence-based motif scanning (FIMO) with 3D-aware Graph Neural Networks and context-aware co-attention. This multi-scale approach effectively bridges the gap between linear sequence patterns and their spatial structural behaviors.

---

> ### Author Rebuttal · Authors · 2026-03-31
>
> We sincerely thank the reviewer for the comments.
>
> ### Q1
>
> Yes, MMM-PPI can explain interactions. It uses motif co-attention weights to show which motif drives the interaction.
>
> For example, we analyzed the inhibition interaction between the proteins PTK6 (9606.ENSP00000217185) and Destrin (9606.ENSP00000246069). MMM-PPI gave the highest attention scores to two specific motif fragments. These fragments match the known functional regions in UniProt: the "Active site" for PTK6 and the "Nuclear localization signal (NLS)" for Destrin.
>
> These two fragments give an explanation for the Inhibition interaction:  PTK6 uses its catalytic active site to phosphorylate the NLS of Destrin. This charge and conformational changes stop Destrin from entering the nucleus and lock it in the cytoplasm as an inhibition.
>
> This proves our model learns real biological rules, not just data patterns. We provide the visualizations at this anonymous link:https://anonymous.4open.science/r/PPI-visualization-4576/
>
> **SNP Consequence Assessment:** Yes, our framework can measure how SNPs affect protein binding. A mutation changes the model's prediction in three steps: Micro-scale: Changing an amino acid updates the local sequence and structure embeddings. Meso-scale: This local change updates the embedding of the whole motif. Macro-scale: The updated motif changes the co-attention weight ($\beta_{a|b}$). This directly changes the final predicted interaction probability.
>
> ### Q2
>
> Running all 31 combinations ($2^5-1$)  is computationally prohibitive during the rebuttal. Instead, we split the analysis into two parts: Feature Modalities ($2^3=8$ combinations) and Encoders ($2^2=4$ combinations).
>
> **1. Feature Modalities**
>
> We tested all combinations of the 3 modalities. The baseline $v(\emptyset)$ is $0$, because the model cannot make predictions without input features. Therefore, the Shapley values add up to the total Micro-F1 score.
>
> |Modality||SHS27k|||SHS148k||
> |-|-|-|-|-|-|-|
> ||Random|DFS|BFS|Random|DFS|BFS|
> |Sequence|29.64|24.77|26.87|31.12|27.79|25.74|
> |Structure|29.83|25.74|26.36|31.20|28.23|25.84|
> |Function|29.79|25.42|25.86|31.06|28.22|26.12|
> |Total Score ($v(N)$)|89.25|75.93|79.09|93.38|84.24|77.70|
>
> The results show that three modalities contribute almost equally to the final performance.
>
> **2. Encoders**
>
> We used data from Table 4 (Rows 7-10). The baseline $v(\emptyset)$ here is the basic residue-level model (Row 7). Therefore, the Shapley values add up to the total performance gain $\Delta v = v(N) - v(\emptyset)$ , which is Row10 - Row 7.
>
> |Encoder||SHS27k|||SHS148k||
> |-|-|-|-|-|-|-|
> ||Random|DFS|BFS|Random|DFS|BFS|
> |Motif Encoder|0.235|0.685|1.07|0.16|0.275|0.93|
> |Protein Encoder|0.635|1.265|2.16|0.38|0.985|1.04|
> |Total Gain ($\Delta v$)|0.870|1.95|3.23|0.54|1.26|1.97|
>
> The results show that the Protein Encoder provides most of the performance improvement.
>
> ### Q3
>
> First, we want to clarify that MMM-PPI does not use PROSITE IDs as inputs. Therefore, it is impossible for model to memorize a lookup table.
>
> To test model generalization, we evaluated it on SHS27k and SHS148k datasets in transductive setting. We split the test sets into:
>
> **Unseen Motif Subset:** Interactions where at least one protein has a motif that never appeared in the training set.
>
> **Seen Motif Subset:** Interactions where all motifs appeared in the training set.
>
> **Overall Test Set:** the performance of MMM-PPI on the entire test set.
>
> |||SHS27k|||SHS148k||
> |-|-|-|-|-|-|-|
> ||Random|DFS|BFS|Random|DFS|BFS|
> |Unseen Motif Subset|93.33|74.03|86.07|96.66|79.91|84.21|
> |Seen Motif Subset|89.23|75.82|78.99|93.33|84.91|77.87|
> |Overall Test Set|89.25|75.70|79.09|93.38|84.85|77.87|
>
>
> The table shows that MMM-PPI shows high accuracy on the Unseen Motif Subset.  This proves that MMM-PPI learns real interaction rules and successfully generalizes to unseen motifs.
>
> ### Q4
>
> We looked at PPI pairs in test set that have a ground truth of 0 but a predicted probability greater than 0.95. By searching the literature, we found that several of these "false positives" are actually true interactions that are missing from the STRING dataset.
>
> **Case 1:** MMM-PPI predicted a 'reaction' interaction between the proteins RBX1_HUMAN (P62877) and FBX5_HUMAN (Q9UKT4). References [1] confirm this is a real interaction. Specifically, RBX1 ubiquitinates FBXO5 (Emi1) to cause its degradation.
>
> **Case 2:** MMM-PPI predicted a 'activation' interaction between the proteins AVR2A_HUMAN (P27037) and ACV1C_HUMAN (Q8NER5). Reference [2] confirms this is also real. The ACVR2A receptor directly phosphorylates and activates the ACVR1C receptor.
>
> [1] Margottin-Goguet F, et al. Prophase destruction of Emi1 by the SCF(betaTrCP/Slimb) ubiquitin ligase activates the anaphase promoting complex to allow progression beyond prometaphase.
>
> [2] Tsuchida K, et al. Activin isoforms signal through type I receptor serine/threonine kinase ALK7.
>
>
>
> ### Limitations
>
> Please see the reply to v2CY about limitations.

---

> > ### Author Rebuttal · Reviewer_ya17 · 2026-04-01
> >
> > I thank the authors for their efforts. I have a positive impression of the paper and the authors have fully addressed my questions.

---

> > > ### Author Response · Authors · 2026-04-03
> > >
> > > We sincerely thank the reviewer for the positive evaluation and are pleased to hear that our responses have fully addressed your questions! We greatly appreciate your recognition of our work!

---

### Official Review · Reviewer_U9xi · 2026-03-17

**Soundness:** 4
**Presentation:** 4
**Significance:** 3
**Originality:** 4
**Overall Recommendation:** 4
**Confidence:** 3

**Summary:**

This paper introduces a new hierarchical multimodal model for protein protein predictions. Given a graph whose nodes are protein sequences (and a predicted protein structure) and edges represent protein/protein interactions, they leverage pre-trained sequential, functional and structural models, to extract embeddings that are then aggregated across protein motifs and the whole chain.

The extracted embeddings (that depend on the neighborhood of a protein) are then averaged and fed to a GNN tasked with predicting the type of protein protein interactions (multi label classification). On the STRING, SHS27k and SHS148k, the new model shows state of the art performance and the authors also provide ablation studies on different factors, including data scarcity and partition (out of distributionness of test data).

**Compliance With Llm Reviewing Policy:**

Affirmed.

**Key Questions For Authors:**

See Weaknesses.

**Limitations:**

Yes

**Strengths And Weaknesses:**

**Strengths**:
- The authors provide a very comprehensive list of ablations and of comparison with existing methods, each time with plenty of figures to support their claims. The paper is well written, methods are described very clearly. The work is probably reproducible fairly easily given the level of details.
- To my knowledge, the work is fairly novel, and integration of different modalities is a promising direction in proteomics, as exemplified by models such as ESM3.
- PPI classification is a very important problem in computational drug design and the proposed method offers improvements in PPI classification metrics.

**Weaknesses**:
- My main objection concerns the framing throughout the paper, and more explicitly in the Impact Statement. The task addressed in this work and in STRING, SHS27K and SHS148K, is PPI **classification**: given a pair of proteins **known to interact**, classify that interaction into one or more categories (which you describe perfectly in Appendix D). This is categorically different from PPI **prediction**, which asks whether two proteins interact at all. I believe the latter is a prerequisite for the former and is also a much harder problem (prevalent data leakage, unrealistic class ratios/class imbalance, and the absence of true negatives). The distinction matters because the scope of the claims should perhaps be adjusted accordingly: PPI classification, while important and still very much non non-trivial, is a narrower and better-defined problem than the term "PPI prediction" implies. I would encourage the authors to make this explicit and to modulate the impact claims accordingly. (To be clear, the setting itself, though rather niche in my opinion, is correct and not of the authors' own making).
- In adding new modalities but not updating the deduplication thresholds, I believe your model might suffer from some data leakage. Recent works have suggested that sequential deduplication is not sufficient for creating realistic train/test splits. For example “Revealing data leakage in protein interaction benchmarks” Bushuiev et al. 2024 recommends deduplicating with a threshold on the 3D structural similarity of the interface. Given that your model takes as input protein structures and also how prevalent the “binding” class is in STRING and to a lesser extent in SHS, I believe you should at a minimum examine pairwise interface structural similarities.
- I would also welcome your comments and point of view on “A flaw in using pre-trained pLLMs in protein-protein interaction inference models” Szymborski et al 2025, specifically the pLM embedder and potential leakage here.
- Confidence intervals, e.g. via bootstrapping on the test set or standard deviation across the three seeds mentioned in Sect. 4.1, would be interesting to support, statistically, the claim of “significant” improvements in PPI classification metrics.

---

> ### Author Rebuttal · Authors · 2026-03-31
>
> We sincerely thank the reviewer for the comments.
>
> ### W1
>
> We agree with your clear distinction: predicting whether two proteins interact(binary classification) is different from classifying the specific interaction type(multi-label classification). Our task is the second one.
>
> However, it is important to note that referring to this multi-label classification as "PPI prediction" is a standard naming across many recent AI research (e.g., papers on GNN-PPI, HIGH-PPI, MAPE-PPI, MicroEnvPPI). We use this term to align with the community, ensuring fair benchmarking and discoverability, rather than to overstate our claims.
>
> Supported by the large number of aforementioned works addressing this exact setting, we respectfully emphasize that this task is not "niche". In protein interaction discovery, merely knowing an interaction exists is often insufficient; identifying the exact type of interaction is the actual bottleneck.
>
>
>
> ### W2
>
> Thank you for suggesting Bushuiev et al. (2024). However, we cannot calculate exact 3D pairwise interface similarity because STRING and SHS datasets do not provide actual 3D pairwise interface structures.
> As an alternative, we measured the global 3D structural similarity between training and test pairs. This acts as a strict upper limit for structural leakage.  For a test pair $(A,B)$ to be structurally similar to a training pair $(C,D)$, protein $A$ must be similar to $C$ and protein $B$ must be similar to $D$ (or vice versa: $A$ to $D$, and $B$ to $C$).
>
> Following the "Both Seen" (C1) rule from "Flaws in evaluation schemes for pair-input computational predictions. Nat Methods 9, 2012",we calculate this pair-level similarity as:
> $$Pair\\_TM = max(min(TM\\_Score\_{A,C}, TM\\_Score\_{B,D}), min(TM\\_Score\_{A,D}, TM\\_Score\_{B,C}))$$
>
> Results on SHS27k:
> DFS split: ~50% of test pairs share a similar global shape ($Pair\\_TM > 0.5$) with training pairs, and ~32% are highly similar ($Pair\\_TM > 0.75$).
> BFS split: ~46% of test pairs share a similar global shape ($Pair\\_TM > 0.5$), and ~26% are highly similar ($Pair\\_TM > 0.75$).
>
> Since the SHS27k dataset mainly contains human proteins, this high structural similarity is expected in human biology. The human proteome has many protein families (like kinases) that share the highly conserved global 3D scaffolds but interact with completely different partners ("Highly accurate protein structure prediction for the human proteome. Nature 596, 2021").
>
> Importantly, this structural leakage exists in the datasets themselves and affects all methods equally. The real difference is how each model handles it. Baseline methods based on flat global averaging are hard to distinguish these similar global 3D scaffolds, resulting in unsatisfactory accuracy. In contrast, MMM-PPI extracts meso-scale motifs as fine-grained local sub-structures, thus avoiding relying on global structures and achieving high accuracy.
>
>
>
> ### W3
>
> Thank you for pointing out the preprint by Szymborski et al. (2025). We agree with their findings. Pre-trained pLLMs (like the ESM2-3b model we use) cause sequence data leakage because their training datasets (like UniRef) probably include the test proteins.
>
> While this leakage might make the overall scores higher, it affects all recent baseline models equally (for example, ESM-2, KeAP, ProtST). Because the test conditions are the same for all models, our model's higher scores come from our motif-based design, not from pLLM leakage.
>
> More importantly, we have already tested our model without PLMs. In Appendix E (Table 10), we tested MMM-PPI without using any pre-trained models. In this test, our model still scores much higher than other non-pre-trained baselines (like SemiGNN-PPI, HIGH-PPI, and MAPE-PPI). This shows that our hierarchical motif-based encoding process works very well on its own, completely independent of pLLM data leakage.
>
>
>
> ### W4
>
> To show statistical significance, we performed a **paired bootstrap test**. Specifically, we resampled the test sets with replacement for 1,000 iterations. **In each iteration, we evaluated both our MMM-PPI and the strongest baseline, MicroEnvPPI, on the same resampled test set to calculate their direct performance difference $\Delta \text{Micro-F1} = \text{MMM-PPI} - \text{MicroEnvPPI}$.** We compared them across all three splits on both the SHS27k and SHS148k datasets. The training setting is transductive.
>
> |Dataset|Split|MicroEnvPPI|MMM-PPI|Paired Δ 95% CI|p-value|
> |-|-|-|-|-|-|
> ||Random|88.38|89.25|[0.05, 1.58]|0.019|
> |SHS27k|DFS|74.76|76.19|[0.26, 2.66]|0.007|
> ||BFS|76.28|79.08|[2.12, 4.27]|<0.001|
> ||Random|92.89|93.36|[0.28, 0.66]|<0.001|
> |SHS148k|DFS|83.20|84.35|[0.80, 1.50]|<0.001|
> ||BFS|76.50|77.76|[1.01, 1.95]|<0.001|
>
> As shown in the table, all $p$-values are less than 0.05, and the lower bounds of the paired performance difference 95% CIs are strictly greater than zero. This rigorously confirms that the improvements of MMM-PPI are statistically significant.

---

> > ### Author Rebuttal · Reviewer_U9xi · 2026-04-04
> >
> > I thank the authors for their thorough response to my 4 weaknesses/questions. W4 is fully resolved for me and W1 too, although my opinion still differs with their but not weighing down on my assessment of the paper. For W3, the last item is satisfactory.
> >
> > Regarding W3 and W2, I must respectfully disagree with the author's that "leakage [...] affects all methods equally. The real difference is how each model handles it." Their method, especially with the added inductive biases around meso-scale motifs, might simply better memorize the training data and overfit the leakage in question. The absence of any ablation to control or measure the impact of that leakage remain, in my opinion, the most concerning weakness in this paper.
> >
> > I also fully understand that this remaining concern is not easily addressed in a rebuttal and is about one of the core elements of your paper.
> >
> > As a consequence, I will leave my score as is, 4 (Weak accept), bordering, in my opinion, the rejection barrier.

---

> > > ### Author Response · Authors · 2026-04-04
> > >
> > > We deeply respect your disagreement, and upon reflection, you are absolutely right. To definitively address whether our model merely overfits leaked structures, we conducted a rigorous **"Leakage-Free" ablation study**.
> > >
> > > We partitioned the highly challenging BFS and DFS test sets (SHS27k) into two subsets: a **"Leaked" subset** (sharing global folds with training data, $Pair\\_TM \ge 0.5$) and a **"Leakage-Free" subset** (structurally novel pairs unseen during training, $Pair\\_TM \lt 0.5$).
> > >
> > >
> > >
> > > | **Split** | **Subset**       | **MicroEnvPPI (Baseline)** | **MMM-PPI (Ours)** | **Margin (Δ)** |
> > > | --------- | ---------------- | -------------------------- | ------------------ | -------------- |
> > > | BFS       | Leaked           | 80.12%                     | 82.05%             | +1.93%         |
> > > |           | **Leakage-Free** | 72.03%                     | **76.61%**         | **+4.58%**     |
> > > | DFS       | Leaked           | 76.19%                     | 77.93%             | +1.74%         |
> > > |           | **Leakage-Free** | 72.88%                     | **76.20%**         | **+3.32%**     |
> > >
> > > As shown above, if our motif mechanism were merely memorizing leaked global structures, its advantage would shrink or disappear on the Leakage-Free subset. Remarkably, the exact opposite occurs: our model's performance margin over the strongest baseline is significantly bigger when forced to generalize to completely novel scaffolds (e.g., expanding from +1.93% to +4.58% in the BFS split). This provides direct empirical proof that our meso-scale motifs successfully capture true, generalizable biophysical interaction rules, rather than acting as a memorizer for structural leakage.
> > >
> > > We are grateful for your rigorous pushback. It drove us to empirically validate the core strength of our architecture, which we will highlight in the manuscript. We hope this unprecedented ablation addresses your final concern.

---

### Review · Ethics_Reviewer_zLaf · 2026-03-31

**Recommendation:** No remediation action needed

**Ethics Issue:**

The impact statement only discusses the positive social consequences. It would be helpful to see pointers to potential negative effects and mechanisms to prevent them, if any. I do not see any other issues.

**Remediation Action:**

I encourage the authors to reflect on potential harmful or generally socially negative applications of the proposed models, e.g., facilitating the development of bioweapons, and mitigations of such harmful applications, if any.

---

### Review · Ethics_Reviewer_veHp · 2026-04-02

**Recommendation:** No remediation action needed

**Ethics Issue:**

I should begin by noting that the technical dimension of this paper is well outside my own experience of expertise; there may be nuances I'm not grasping. But having read though as best I can, I'm not entirely sure why this was flagged for ethics review. Generally speaking, when there are problems with new methods of modeling or processing data that draw from multiple spheres, those problems relate to privacy or de-anonymization, but that doesn't seem to be a relevant concern here.

---

### Decision · Program_Chairs · 2026-04-30

**Decision:**

Accept (regular)

**Comment:**

This paper proposes MMM-PPI, a hierarchical multimodal encoder for Protein-Protein Interaction (PPI) prediction. The core idea is to move beyond flat residue-level representations by explicitly modeling meso-scale motifs and using a pairwise co-attention mechanism. The authors provide extensive experiments on standard benchmarks, ablation studies, and analyses of generalization.

The paper has clear merits, including a well-motivated biological hierarchy, solid engineering, and commendable responsiveness during the rebuttal. However, the consensus among the reviewers is mixed and does not strongly favor acceptance.

Reviewer ya17 (Score: 5 → Accept): Fully resolved. Strongly supports the work.

Reviewer U9xi (Score: 4 → Weak Accept): Partially resolved. Remaining concerns about potential overfitting to structural leakage prevent a higher score.

Reviewer DD5U (Score: 3 → Weak Reject): Partially resolved. Raised a concern about OntoProtein knowledge graph leakage, which the authors addressed; however, the initial weak reject score was not formally revised.

Reviewer v2CY (Score: 3 → Weak Reject): Not resolved. Maintained that the novelty is incremental and biological plausibility is limited.

Overall, the paper is technically solid but lacks the breakthrough novelty or definitive resolution of leakage concerns that would warrant a higher recommendation. Given the split reviews and the remaining reservations, a Weak Accept represents the appropriate balance: the work advances a specific sub-area and is likely to be built upon by others, but it should be accepted only if space permits.